# ACCELERATED INFORMATION GRADIENT FLOW

## ABSTRACT

We present a systematic framework for the Nesterov's accelerated gradient flows in the spaces of probabilities embedded with information metrics. Here two metrics are considered, including both the Fisher-Rao metric and the Wasserstein-2 metric. For the Wasserstein-2 metric case, we prove the convergence properties of the accelerated gradient flows, and introduce their formulations in Gaussian families. Furthermore, we propose a practical discrete-time algorithm in particle implementations with an adaptive restart technique. We formulate a novel bandwidth selection method, which learns the Wasserstein-2 gradient direction from Brownian-motion samples. Experimental results including Bayesian inference show the strength of the current method compared with the state-of-the-art.

## 1 INTRODUCTION

Recently, optimization problems on the space of probability and probability models attract increasing attentions from machine learning communities. These problems include variational inference (Blei et al., 2017), Bayesian inference (Liu & Wang, 2016), Generative Adversary Networks (GAN, Goodfellow et al. (2014)), and policy optimizations (Zhang et al., 2018), etc. For instance, variational inference methods approximate a target density by minimizing the Kullback-Leibler (KL) divergence as the loss (objective) function.

Gradient descent methods with sampling efficient properties play essential roles to solve these optimization problems. Here the gradient descent direction often relies on the information metric over the probability space. This direction naturally reflects the change of the loss function with respect to the metric. In literature, two important information metrics, such as the Fisher-Rao metric and the Wasserstein-2 (in short, Wasserstein) metric, are of great interests (Amari, 1998; Otto, 2001; Lafferty, 1988). For the Fisher-Rao gradient, classical results including Adam (Kingma & Ba, 2014) and K-FAC (Martens & Grosse, 2015) demonstrate its effectiveness in probability models. For the Wasserstein gradient, many classical methods such as Markov chain Monte Carlo (MCMC) methods (Geman & Geman, 1987; Neal et al., 2011; Welling & Teh, 2011) and particle-based variational inference (ParVI) methods (Liu & Wang, 2016; Chen & Zhang, 2017; Chen et al., 2018) are based on this framework in the probability space. The strength of using the Wasserstein gradient is also shown in probability models such as GANs. (Arjovsky et al., 2017; Lin et al., 2018; Li et al., 2019).

The Nesterov's accelerated method (Nesterov, 1983) is widely applied in accelerating the vanilla gradient descent under the Euclidean metric. It corresponds to a damped Hamiltonian flow, known as the accelerated gradient flow (Su et al., 2016). A natural question is *whether there exists a counterpart of the accelerated gradient flow in the probability space under information metrics*. For optimization problems on a Riemannian manifold, the accelerated gradient methods are studied by Liu et al. (2017); Zhang & Sra (2018). The probability space embedded with information metric can be viewed as a Riemannian manifold. Several previous works explore accelerated methods in this manifold under the Wasserstein metric. Liu et al. (2018; 2019) propose an acceleration framework of ParVI methods based on manifold optimization. Taghvaei & Mehta (2019) introduce the accelerated flow from an optimal control perspective. On the other hand, Cheng et al. (2017); Ma et al. (2019) explore and analyze the acceleration on MCMC, based on the underdamped Langevin dynamics.

In this paper, we present a unified framework of accelerated gradient flows in the probability space embedded with information metrics, named Accelerated Information Gradient (AIG) flows. From an information-differential-geometry perspective, we derive AIG flows by damping Hamiltonian flows, concerning both the Fisher-Rao metric and the Wasserstein metric. Then we focus on the Wasser-

stein metric with the KL divergence loss function. In Gaussian families, we verify the existence of the solution to AIG flows. Here we show that the AIG flow corresponds to a well-posed ODE system in the space of symmetric positive definite matrices. We rigorously prove the convergence rate of AIG flows based on the geodesic convexity of the loss function. Here we note that our proof removes the unnecessary technical assumption in (Taghvaei & Mehta, 2019, Theorem 1).

Besides, we handle two difficulties in numerical implementations of AIG flows. On the one hand, as pointed out by Taghvaei & Mehta (2019); Liu et al. (2019), the logarithm of density term (Wasserstein gradient of KL divergence) is hard to approximate in particle formulations. We propose a novel kernel selection method, whose bandwidth is learned by sampling from Brownian motions. We call it the BM method. On the other hand, we notice that the AIG flow can be a numerically stiff system, especially in high-dimensional sample spaces. This is because the solution of AIG flows can be close to the boundary of the probability space. To handle this issue, we propose an adaptive restart technique, which accelerates and stabilizes the AIG algorithm. Numerical results in toy examples, Gaussian measures and Bayesian Logistic regression indicate the validity of the BM method and the acceleration effects of the proposed AIG flow.

This paper is organized as follows. Section 2 briefly reviews the information metrics and their corresponding gradient flows and Hamiltonian flows in the probability space. In Section 3, we formulate various forms of AIG flows and analyze W-AIG flows in Gaussian measures. We theoretically prove the convergence rate of W-AIG flows in Section 4. Section 5 presents the discrete-time algorithm for W-AIG flows, including the BM method and the adaptive restart technique. Section 6 provides numerical experiments.

## 2 METRIC AND FLOWS IN THE PROBABILITY SPACE

Suppose that $\Omega$ is a region in $\mathbb{R}^n$. Let $\mathcal{F}(\Omega)$ denote the set of smooth functions on $\Omega$. $\langle \cdot, \cdot \rangle$ and $\| \cdot \|$ are the Euclidean inner product and norm in $\mathbb{R}^n$. $\nabla$, $\nabla\cdot$ and $\Delta$ represent the gradient, divergence and Laplacian operators in $\mathbb{R}^n$. Denote the set of probability density

$$\mathcal{P}(\Omega) = \left\{ \rho \in \mathcal{F}(\Omega) \colon \int_\Omega \rho dx = 1, \quad \rho \geq 0 \right\}.$$

The tangent space at $\rho \in \mathcal{P}(\Omega)$ follows $T_\rho(\Omega) = \left\{ \sigma \in \mathcal{F}(\Omega) : \int \sigma dx = 0. \right\}$. The cotangent space at $\rho$, $T_\rho^* \mathcal{P}(\Omega)$, can be treated as the quotient space $\mathcal{F}(\Omega)/\mathbb{R}$, which are functions in $\mathcal{F}(\Omega)$ defined up to addition of constants.

**Definition 1 (Metric in the probability space)** A metric tensor $G(\rho) : T_\rho \mathcal{P}(\Omega) \to T_\rho^* \mathcal{P}(\Omega)$ is an invertible mapping from the tangent space at $\rho$ to the cotangent space at $\rho$. This metric tensor defines the metric (inner product) on the tangent space $T_\rho \mathcal{P}(\Omega)$. Namely, for $\sigma_1, \sigma_2 \in T_\rho \mathcal{P}(\Omega)$, we define

$$g_\rho(\sigma_1, \sigma_2) = \int \sigma_1 G(\rho) \sigma_2 dx = \int \Phi_1 G(\rho)^{-1} \Phi_2 dx,$$

where $\Phi_i$ is the solution to $\sigma_i = G(\rho)^{-1} \Phi_i$, $i = 1, 2$.

We present two important examples of metrics on the probability space $\mathcal{P}(\Omega)$: the Fisher-Rao metric from information geometry and the Wasserstein metric from optimal transport.

**Example 1 (Fisher-Rao metric)** The inverse of the Fisher-Rao metric tensor is defined by

$$G^F(\rho)^{-1} \Phi = \rho \left( \Phi - \int \Phi \rho dx \right), \quad \Phi \in T_\rho^* \mathcal{P}(\Omega).$$

The Fisher-Rao metric on the tangent space is given by

$$g_\rho^F(\sigma_1, \sigma_2) = \int \Phi_1 \Phi_2 \rho dx - \left( \int \Phi_1 \rho dx \right) \left( \int \Phi_2 \rho dx \right), \quad \sigma_1, \sigma_2 \in T_\rho \mathcal{P}(\Omega),$$

where $\Phi_i$ is the solution to $\sigma_i = \rho \left( \Phi_i - \int \Phi_i \rho dx \right)$, $i = 1, 2$.

**Example 2 (Wasserstein metric)** The inverse of the Wasserstein metric tensor is defined by

$$G^W(\rho)^{-1} \Phi = -\nabla \cdot (\rho \nabla \Phi), \quad \Phi \in T_\rho^* \mathcal{P}(\Omega).$$

The Wasserstein metric on the tangent space is given by

$$g_\rho^W(\sigma_1, \sigma_2) = \int \rho \left\langle \nabla\Phi_1, \nabla\Phi_2 \right\rangle dx, \quad \sigma_1, \sigma_2 \in T_\rho \mathcal{P}(\Omega),$$

where $\Phi_i$ is the solution to $\sigma_i = -\nabla \cdot (\rho\nabla\Phi_i)$, $i = 1, 2$.

## 2.1 GRADIENT FLOWS

In learning, many problems can be formulated as the optimization problem in the probability space,

$$\min_{\rho \in \mathcal{P}(\Omega)} E(\rho).$$

Here $E(\rho)$ is a divergence or metric loss functional between $\rho$ and a target density $\rho^* \in \mathcal{P}(\Omega)$. One typical example of $E(\rho)$ is the KL divergence from $\rho$ to $\rho^*$,

$$E(\rho) = D_{\text{KL}}(\rho\|\rho^*) = \int \log\left(\frac{\rho}{\rho^*}\right)\rho dx.$$

Another example is the maximum mean discrepancy (MMD, Gretton et al. (2012)),

$$E(\rho) = \text{MMD}(\rho, \rho^*) = \int\int (\rho(x) - \rho^*(x))K(x, y)(\rho(y) - \rho^*(y))dxdy,$$

where $K(x, y)$ is a given kernel function. The gradient flow for $E(\rho)$ in $(\mathcal{P}(\Omega), g_\rho)$ takes the form

$$\partial_t \rho_t = -G(\rho_t)^{-1}\frac{\delta E(\rho_t)}{\delta\rho_t}.$$

Here $\frac{\delta E(\rho_t)}{\delta\rho_t}$ is the $L^2$ first variation w.r.t. $\rho_t$. We formulate the gradient flow under either the Fisher-Rao metric or the Wasserstein metric.

**Example 3 (Fisher-Rao gradient flow)** The Fisher-Rao gradient flow is given by

$$\partial_t\rho_t = -\rho_t\left(\frac{\delta E}{\delta\rho_t} - \int \frac{\delta E}{\delta\rho_t}\rho_t dx\right).$$

**Example 4 (Wasserstein gradient flow)** The Wasserstein gradient flow writes

$$\partial_t\rho_t = \nabla \cdot \left(\rho_t\nabla\frac{\delta E}{\delta\rho_t}\right).$$

## 2.2 HAMILTONIAN FLOW

In this subsection, we briefly review the Hamiltonian flow in the probability space. By using the metric $g_\rho$ in the probability space, we can define a Lagrangian by

$$\mathcal{L}(\rho_t, \partial_t\rho_t) = \frac{1}{2}g_{\rho_t}(\partial_t\rho_t, \partial_t\rho_t) - E(\rho_t).$$

The Euler-Lagrange equation for the Lagrangian follows

$$\partial_t\left(\frac{\delta\mathcal{L}}{\delta(\partial_t\rho_t)}\right) = \frac{\delta\mathcal{L}}{\delta\rho_t} + C(t), \tag{1}$$

where $C(t)$ is a spatially-constant function.

**Proposition 1** *If we let $\Phi_t = \delta L/\delta(\partial_t\rho_t) = G(\rho_t)\partial_t\rho_t$, the equation (1) can be formulated as a system of $(\rho_t, \Phi_t)$, i.e.,*

$$\begin{cases} \partial_t\rho_t - G(\rho_t)^{-1}\Phi_t = 0, \\ \partial_t\Phi_t + \frac{1}{2}\frac{\delta}{\delta\rho_t}\left(\int \Phi_t G(\rho_t)^{-1}\Phi_t dx\right) + \frac{\delta E}{\delta\rho_t} = 0, \end{cases} \tag{2}$$

*where $\Phi_t$ is up to a spatially-constant function shrift. Here (2) is the Hamiltonian flow*

$$\partial_t\begin{bmatrix} \rho_t \\ \Phi_t \end{bmatrix} - \begin{bmatrix} 0 & 1 \\ -1 & 0 \end{bmatrix}\begin{bmatrix} \frac{\delta}{\delta\rho_t}\mathcal{H}(\rho_t, \Phi_t) \\ \frac{\delta}{\delta\Phi_t}\mathcal{H}(\rho_t, \Phi_t) \end{bmatrix} = 0, \tag{3}$$

*with respect to the Hamiltonian $\mathcal{H}(\rho_t, \Phi_t) = \frac{1}{2}\int \Phi_t G(\rho_t)^{-1}\Phi_t dx + E(\rho_t)$.*

Similarly, we can write the Hamiltonian flow under the Fisher-Rao metric or the Wasserstein metric.

**Example 5 (Fisher-Rao Hamiltonian flow)** The Fisher-Rao Hamiltonian flow follows

$$
\begin{cases}
\partial_t \rho_t - \Phi_t \rho_t + \int \Phi_t \rho_t dx = 0, \\
\partial_t \Phi_t + \dfrac{1}{2} \Phi_t^2 - \left( \int \rho_t \Phi_t dx \right) \Phi_t + \dfrac{\delta E}{\delta \rho_t} = 0.
\end{cases}
$$

The corresponding Hamiltonian is $\mathcal{H}^F(\rho_t, \Phi_t) = \frac{1}{2} \left( \int \Phi_t^2 \rho_t dx - \left( \int \rho_t \Phi_t dx \right)^2 \right) + E(\rho_t)$.

**Example 6 (Wasserstein Hamiltonian flow)** The Wasserstein Hamiltonian flow writes

$$
\begin{cases}
\partial_t \rho_t + \nabla \cdot (\rho_t \nabla \Phi_t) = 0, \\
\partial_t \Phi_t + \dfrac{1}{2} \|\nabla \Phi_t\|^2 + \dfrac{\delta E}{\delta \rho_t} = 0.
\end{cases}
$$

The corresponding Hamiltonian is $\mathcal{H}^W(\rho_t, \Phi_t) = \frac{1}{2} \int \|\nabla \Phi_t\|^2 \rho_t dx + E(\rho_t)$. This is identical to the Wasserstein Hamiltonian flow introduced by Chow et al. (2019).

## 3 ACCELERATED INFORMATION GRADIENT FLOW

Let $\alpha_t \geq 0$ be a scalar function of $t$. We add a damping term $\alpha_t \Phi_t$ to the Hamiltonian flow (3).

$$
\partial_t \begin{bmatrix} \rho_t \\ \Phi_t \end{bmatrix} + \begin{bmatrix} 0 \\ \alpha_t \Phi_t \end{bmatrix} - \begin{bmatrix} 0 & 1 \\ -1 & 0 \end{bmatrix} \begin{bmatrix} \frac{\delta}{\delta \rho_t} \mathcal{H}(\rho_t, \Phi_t) \\ \frac{\delta}{\delta \Phi_t} \mathcal{H}(\rho_t, \Phi_t) \end{bmatrix} = 0. \tag{4}
$$

This renders the Accelerated Information Gradient (AIG) flow

$$
\begin{cases}
\partial_t \rho_t - G(\rho_t)^{-1} \Phi_t = 0, \\
\partial_t \Phi_t + \alpha_t \Phi_t + \dfrac{1}{2} \dfrac{\delta}{\delta \rho_t} \left( \int \Phi_t G(\rho_t)^{-1} \Phi_t dx \right) + \dfrac{\delta E}{\delta \rho_t} = 0,
\end{cases} \tag{AIG}
$$

with initial values $\rho_0 = \rho^0$ and $\Phi_0 = 0$. The choice of $\alpha_t$ depends on the geodesic convexity of $E(\rho)$, which has an equivalent definition as follows.

**Definition 2** For a functional $E(\rho)$ defined on the probability space, we say that $E(\rho)$ has a $\beta$-positive Hessian (in short, Hess($\beta$)) w.r.t. the metric $g_\rho$ if there exists a constant $\beta \geq 0$ such that for any $\rho \in \mathcal{P}(\Omega)$ and any $\sigma \in T_\rho \mathcal{P}(\Omega)$, we have

$$
g_\rho(\text{Hess } E(\rho)\sigma, \sigma) \geq \beta g_\rho(\sigma, \sigma).
$$

Here Hess is the Hessian operator w.r.t. $g_\rho$.

If $E(\rho)$ is Hess($\beta$) for $\beta > 0$, then $\alpha_t = 2\sqrt{\beta}$; if $E(\rho)$ is Hess(0), then $\alpha_t = 3/t$.

**Remark 1** The Nesterov's accelerated method (Nesterov, 1983) is a first-order method to optimize $f(x)$ in the Euclidean space. The corresponding accelerated gradient flow by Su et al. (2016) is equivalent to a damped Hamiltonian system

$$
\begin{bmatrix} \dot{x} \\ \dot{p} \end{bmatrix} + \begin{bmatrix} 0 \\ \alpha_t p \end{bmatrix} - \begin{bmatrix} 0 & I \\ -I & 0 \end{bmatrix} \begin{bmatrix} \nabla_x H^E(x, p) \\ \nabla_p H^E(x, p) \end{bmatrix} = 0, \quad H^E(x, p) = \dfrac{1}{2} \|p\|^2 + f(x).
$$

with initial values $x(0) = x_0$ and $p(0) = 0$. The choice of $\alpha_t$ depends on the property of $f(x)$. If $f(x)$ is $\beta$-strongly convex, then $\alpha_t = 2\sqrt{\beta}$; if $f(x)$ is convex, then $\alpha_t = 3/t$. We apply this Hamiltonian flow interpretation to construct (AIG) in the probability space with information metrics.

We give examples of AIG flows under either the Fisher-Rao metric or the Wasserstein metric.

**Example 7 (Fisher-Rao AIG flow)** The Fisher-Rao AIG flow writes

$$
\begin{cases}
\partial_t \rho_t - \Phi_t \rho_t + \left( \int \Phi_t \rho_t dx \right) \rho_t = 0, \\
\partial_t \Phi_t + \alpha_t \Phi_t + \dfrac{1}{2} \Phi_t^2 - \left( \int \rho_t \Phi_t dx \right) \Phi_t + \dfrac{\delta E}{\delta \rho_t} = 0.
\end{cases} \tag{F-AIG}
$$

**Example 8 (Wasserstein AIG flow)** The Wasserstein AIG flow writes

$$\begin{cases} \partial_t \rho_t + \nabla \cdot (\rho_t \nabla \Phi_t) = 0, \\ \partial_t \Phi_t + \alpha_t \Phi_t + \frac{1}{2} \|\nabla \Phi_t\|^2 + \frac{\delta E}{\delta \rho_t} = 0. \end{cases} \quad \text{(W-AIG)}$$

For the rest of this paper, we mainly focus on the Wasserstein metric. Here the AIG flow (Eulerian formulation in fluid dynamics) has a counterpart in the particle level (Lagrangian formulation).

**Proposition 2** *Suppose that $X_t \sim \rho_t$ and $V_t = \nabla \Phi_t(X_t)$ are the position and the velocity of a particle at time $t$. Then, the differential equation of the particle system corresponding to* (W-AIG) *writes*

$$dX_t = V_t dt, \quad dV_t = -\alpha_t V_t dt - \nabla \left( \frac{\delta E}{\delta \rho_t} \right)(X_t) dt. \quad \text{(W-AIG-P)}$$

If $E(\rho)$ evaluates the KL divergence and $\rho^* \propto \exp(-f(x))$, (W-AIG-P) is equivalent to

$$dX_t = V_t dt, \quad dV_t = -\alpha_t V_t dt - \nabla f(X_t) dt - \nabla \log \rho_t(X_t) dt. \quad \text{(W-AIG-P-KL)}$$

**Remark 2** The $\nabla \log \rho_t(X_t) dt$ term cannot be simply replaced by a Brownian motion $dB_t$ because $\rho_t$ is the marginal distribution on $X_t$. Several previous works have studied the accelerated gradient flow of KL divergence in the probability space under the Wasserstein metric. Taghvaei & Mehta (2019) construct the accelerated gradient flow in the probability space based on Wibisono et al. (2016)'s variational formulation on the Nesterov's accelerated method. Their flows coincide with (W-AIG-P-KL) with $\alpha_t = 3/t$ after rescaling. The underdamped Langevin dynamics in (Cheng et al., 2017; Ma et al., 2019) damps the Hamiltonian flow of the particles, which is different from (W-AIG-P) as shown in (Taghvaei & Mehta, 2019). Liu et al. (2018; 2019) give the discrete-time accelerated algorithm from the perspective of manifold optimization.

## 3.1 WASSERSTEIN METRIC RESTRICTED TO GAUSSIAN

In this subsection, we demonstrate that (W-AIG) has an ODE formulation in Gaussian family. Denote $\mathcal{N}_n^0$ to the multivariate Gaussian densities with zero means. Namely, if $\rho^0, \rho^* \in \mathcal{N}_n^0$, then we show that (W-AIG) has a solution $(\rho_t, \Phi_t)$ and $\rho_t \in \mathcal{N}_n^0$.

Let $\mathbb{P}^n$ and $\mathbb{S}^n$ represent symmetric positive definite matrix and symmetric matrix with size $n \times n$ respectively. Each $\rho \in \mathcal{N}_n^0$ can be uniquely expressed by its covariance matrix $\Sigma \in \mathbb{P}^n$ by $\rho(x; \Sigma) = \frac{(2\pi)^{-n/2}}{\sqrt{\det(\Sigma)}} \exp\left(-\frac{1}{2}x^T \Sigma^{-1} x\right)$. The Wasserstein metric on $\mathcal{P}(\mathbb{R}^n)$ induces a metric on $\mathcal{N}_n^0$, which is a totally-geodesic submanifold in $\mathcal{P}(\mathbb{R}^n)$, see (Takatsu, 2008; Modin, 2016; Malagò et al., 2018). So there exists a Wasserstein metric on $\mathbb{P}^n$, also known as the Bures metric. For $\Sigma \in \mathbb{P}^n$, the tangent space and cotangent space follow $T_\Sigma \mathbb{P}^n \simeq T_\Sigma^* \mathbb{P}^n \simeq \mathbb{S}^n$.

**Definition 3 (Wasserstein metric in Gaussian)** For $\Sigma \in \mathbb{P}^n$, the metric tensor $G(\Sigma) : \mathbb{S}^n \to \mathbb{S}^n$ is defined by $G(\Sigma)^{-1} S = 2(\Sigma S + S\Sigma)$. The Wasserstein metric on $\mathbb{S}^n$ is $g_\Sigma(A_1, A_2) = \text{tr}(A_1 G(\Sigma) A_2) = \text{tr}(S_1 \Sigma S_2)$, where $S_i \in \mathbb{S}^n$ is the solution to $A_i = \Sigma S_i + S_i \Sigma$, $i = 1, 2$.

**Proposition 3** *Given an energy function $E(\Sigma)$, the Wasserstein gradient flow in Gaussian writes*

$$\dot{\Sigma}_t = -2(\Sigma_t \nabla_{\Sigma_t} E(\Sigma_t) + \nabla_{\Sigma_t} E(\Sigma_t) \Sigma_t).$$

*Here $\nabla_{\Sigma_t}$ is the standard matrix derivative. The Hamiltonian flow is a system of $(\Sigma_t, S_t)$, i.e.,*

$$\begin{cases} \dot{\Sigma}_t - (S_t \Sigma_t + \Sigma_t S_t) = 0, \\ \dot{S}_t + S_t^2 + 2\nabla_{\Sigma_t} E(\Sigma_t) = 0. \end{cases} \quad (5)$$

*The corresponding Hamiltonian writes $H(\Sigma_t, S_t) = \text{tr}(S_t \Sigma_t S_t) + 2E(\Sigma_t)$.*

Therefore, by adding the damping term $\alpha_t S_t$, we obtain the Wasserstein AIG flow in Gaussian.

$$\begin{cases} \dot{\Sigma}_t - (S_t \Sigma_t + \Sigma_t S_t) = 0, \\ \dot{S}_t + \alpha_t S_t + S_t^2 + 2\nabla_{\Sigma_t} E(\Sigma_t) = 0, \end{cases} \quad \text{(W-AIG-G)}$$

with initial values $\Sigma_0 = \Sigma^0$ and $S_0 = 0$. For now, we consider $E(\Sigma)$ to be the KL divergence.

$$E(\Sigma) \triangleq E(\rho(\,\cdot\,;\Sigma)) = \frac{1}{2}\left[\operatorname{tr}(\Sigma(\Sigma^*)^{-1}) - \log\det(\Sigma(\Sigma^*)^{-1}) - n\right], \qquad (6)$$

where $\Sigma^*$ is the covariance matrix of $\rho^*$. The following theorem proves the well-posedness of (W-AIG-G) and illustrates the connection between W-AIG flows in $\mathbb{P}^n$ and $\mathcal{P}(\mathbb{R}^n)$.

**Theorem 1** *Suppose that $\rho^0, \rho^* \in \mathcal{N}_0^n$ and their covariance matrices are $\Sigma^0$ and $\Sigma^*$. $E(\Sigma)$ defined in (6) evaluates the KL divergence from $\rho$ to $\rho^*$. Let $(\Sigma_t, S_t)$ be the solution to (W-AIG-G) with initial values $\Sigma_0 = \Sigma^0$ and $S_0 = 0$. Then, for any $t \geq 0$, $\Sigma_t$ is well-defined and stays positive definite. Furthermore, we denote*

$$\rho_t(x) = \frac{(2\pi)^{-n/2}}{\sqrt{\det(\Sigma_t)}}\exp\left(-\frac{1}{2}x^T\Sigma_t^{-1}x\right), \quad \Phi_t(x) = \frac{1}{2}x^T S_t x + C(t),$$

*where $C(t) = -t + \frac{1}{2}\int_0^t \log\det(\Sigma_s(\Sigma^*)^{-1})ds$. Then, $(\rho_t, \Phi_t)$ is the solution to (W-AIG) with initial values $\rho_0 = \rho^0$ and $\Phi_0 = 0$.*

## 4 Convergence rate analysis on W-AIG flows

In this section, we prove the convergence rate of (W-AIG).

**Theorem 2** *Suppose that $E(\rho)$ satisfies Hess($\beta$) for $\beta > 0$. The solution $\rho_t$ to (W-AIG) with $\alpha_t = 2\sqrt{\beta}$ satisfies*

$$E(\rho_t) \leq C_0 e^{-\sqrt{\beta}t} = \mathcal{O}(e^{-\sqrt{\beta}t}).$$

*If $E(\rho)$ only satisfies Hess(0), then the solution $\rho_t$ to (W-AIG) with $\alpha_t = 3/t$ satisfies*

$$E(\rho_t) \leq C_0' t^{-2} = \mathcal{O}(t^{-2}).$$

*Here the constants $C_0, C_0'$ only depend on $\rho_0$.*

**Remark 3** Here Hess($\beta$) is equivalent to the $\beta$-geodesic convexity in the probability space w.r.t. $g_\rho$. For the Wasserstein metric, it is also known as $\beta$-displacement convexity; see (Villani, 2008, Chap 16). Consider the case where $E(\rho)$ is the KL divergence and the target density takes the form $\rho^* \propto \exp(-f(x))$. A sufficient condition for Hess($\beta$) is that $f(x)$ is $\beta$-strongly convex, see (Otto & Villani, 2000; Bakry & Émery, 1985). If $E(\rho)$ satisfies Hess($\beta$) for $\beta > 0$, then the classical analysis indicates that the solution to the Wasserstein gradient flow has an $\mathcal{O}(e^{-2\beta t})$ convergence rate. The W-AIG flow improves the convergence rate to $\mathcal{O}(e^{-\sqrt{\beta}t})$, especially when $\beta$ is close to $0$.

Here we provide a sketch in the proof of Theorem 2. Given $\rho_t$, we can find the optimal transport plan $T_t$ from $\rho_t$ to $\rho^*$. Let $T\#\rho$ denote the push-forward density from $\rho$ by the mapping $T$. The following proposition characterizes the inverse of the exponential map in probability space with the Wasserstein metric.

**Proposition 4** *Denote the geodesic curve $\gamma(s)$ that connects $\rho_t$ and $\rho^*$ by $\gamma(s) = (sT_t + (1 - s)\operatorname{Id})\#\rho_t$, $s \in [0,1]$. Here $\operatorname{Id}$ is the identity mapping from $\mathbb{R}^n$ to itself. Then, $\dot{\gamma}(0)$ corresponds to a tangent vector $-\nabla \cdot (\rho_t(x)(T_t(x) - x)) \in T_{\rho_t}\mathcal{P}(\Omega)$.*

We first consider the case where $E(\rho)$ satisfies Hess($\beta$) for $\beta > 0$. Motivated by the Lyapunov function for Nesterov's ODE in the Euclidean case, we construct the following Lyapunov function.

$$\mathcal{E}(t) = e^{\sqrt{\beta}t}\left(\frac{1}{2}\int\left\|-\sqrt{\beta}(T_t(x) - x) + \nabla\Phi_t(x)\right\|^2 \rho_t(x)dx + E(\rho_t) - E(\rho^*)\right). \qquad (7)$$

**Proposition 5** *Suppose that $E(\rho)$ satisfies Hess($\beta$) for $\beta > 0$. $\rho_t$ is the solution to (W-AIG) with $\alpha_t = 2\sqrt{\beta}$. Then, $\mathcal{E}(t)$ defined in (7) satisfies $\dot{\mathcal{E}}(t) \leq 0$. As a result,*

$$E(\rho_t) \leq e^{-\sqrt{\beta}t}\mathcal{E}(t) \leq e^{-\sqrt{\beta}t}\mathcal{E}(0) = \mathcal{O}(e^{-\sqrt{\beta}t}).$$

Note that $\mathcal{E}(0)$ only depends on $\rho_0$. This proves the first part of Theorem 2. We now consider the case where $E(\rho)$ satisfies Hess(0). We construct the following Lyapunov function.

$$\mathcal{E}(t) = \frac{1}{2} \int \left\| -(T_t(x) - x) + \frac{t}{2} \nabla \Phi_t(x) \right\|^2 \rho_t(x) dx + \frac{t^2}{4} (E(\rho_t) - E(\rho^*)). \tag{8}$$

**Proposition 6** *Suppose that $E(\rho)$ satisfies Hess(0). $\rho_t$ is the solution to* (W-AIG) *with $\alpha_t = 3/t$. Then, $\mathcal{E}(t)$ defined in* (8) *satisfies $\dot{\mathcal{E}}(t) \leq 0$. As a result,*

$$E(\rho_t) \leq \frac{4}{t^2} \mathcal{E}(t) \leq \frac{4}{t^2} \mathcal{E}(0) = \mathcal{O}(t^{-2}).$$

Because $\mathcal{E}(0)$ only depends on $\rho_0$, we complete the proof.

**Remark 4** For the Hess(0) case, we obtain the same result in (Taghvaei & Mehta, 2019, Theorem 1). Their proof comes from the Lagrangian formulation (W-AIG-P) and our proof is based on the Eulerian formulation (W-AIG). However, their technical assumption $\mathbb{E}\left[\left(X_t + e^{-\gamma_t} Y_t - T_{\rho_t}^{\rho_\infty}(X_t)\right) \cdot \frac{d}{dt} T_{\rho_t}^{\rho_\infty}(X_t)\right] = 0$ is only valid in 1-dimensional case. In Appendix C.4, we prove that this quantity is **non-negative**. This is due to the Hodge decomposition behind the optimal transport, see Lemma 1 in Appendix C.3.

## 5 DISCRETE-TIME ALGORITHM FOR W-AIG FLOWS

In this section, we present the discrete-time implementation of (W-AIG-P-KL). This implementation is simpler and more stable than the one in (Taghvaei & Mehta, 2019). Suppose that initial positions of a particle system $\{X_0^i\}_{i=1}^N$ are given and $V_0^i = 0$. The time parameter $t$ is related to the step size $\sqrt{\tau}$ via $t = \sqrt{\tau} k$. The update rule follows

$$V_{k+1}^i = \alpha_k V_k^i - \sqrt{\tau}(\nabla f(X_k^i) + \xi_k(X_k^i)), \quad X_{k+1}^i = X_k^i + \sqrt{\tau} V_{k+1}^i, \tag{9}$$

for $i = 1, 2 \ldots N$. If $E(\rho)$ is Hess($\beta$), then $\alpha_k = \frac{1 - \sqrt{\beta\tau}}{1 + \sqrt{\beta\tau}}$; if $E(\rho)$ is Hess(0) or $\beta$ is unknown, then $\alpha_k = \frac{k-1}{k+2}$. Here $\xi_k(x)$ is an approximation of $\nabla \log \rho_t(x)$.

We review two common choices of $\xi_k$ as follows. If $X_k^i$ follows a Gaussian distribution, then

$$\xi_k(x) = -\Sigma_k^{-1}(x - m_k), \tag{10}$$

where $m_k$ and $\Sigma_k$ are the mean and the covariance matrix of $\{X_k^i\}_{i=1}^N$. For the non-Gaussian case, we use the kernel density estimation (KDE, Singh (1977)), $\tilde{\rho}_k(x) = \frac{1}{N} \sum_{i=1}^N K(x, X_k^i)$ to approximate $\rho_t(x)$. Here $K(x, y)$ is a kernel function. Then, $\xi_k$ writes

$$\xi_k(x) = \nabla \log \tilde{\rho}_k(x) = \frac{\sum_{i=1}^N \nabla_x K(x, X_k^i)}{\sum_{i=1}^N K(x, X_k^i)}. \tag{11}$$

A common choice of $K(x, y)$ is a Gaussian kernel with the bandwidth $h$, $K^G(x, y; h) = (2\pi h)^{-n/2} \exp\left(\|x - y\|^2 / (2h)\right)$. There are two difficulties in the discretization. For one thing, the bandwidth $h$ strongly affects the estimation of $\nabla \log \rho_t$, so we propose the BM method to adaptively learn the bandwidth from samples. For another, the second equation in (W-AIG) is the Hamilton-Jacobi equation, which usually has strong stiffness. We propose an adaptive restart technique to deal with this problem.

**Remark 5** Our numerical implementations of W-AIG flows can be viewed as a special case of ParVI methods. Compared to traditional MCMC methods, ParVI methods are more sample-efficient because make full use of a finite number of particles by taking particle interaction into account.

### 5.1 LEARN THE BANDWIDTH VIA BROWNIAN MOTION

SVGD uses a median (MED) method to choose the bandwidth, i.e.,

$$h_k = \frac{1}{2 \log(N+1)} \text{median}\left(\{\|X_k^i - X_k^j\|^2\}_{i,j=1}^N\right). \tag{12}$$

Liu et al. (2018) propose a Heat Equation (HE) method to adaptively adjust bandwidth. Motivated by the HE method, we introduce the Brownian motion (BM) method to adaptively learn the bandwidth.

Given the bandwidth $h$, $\{X^i\}_{i=1}^N$ and a step size $s$, we can compute two particle systems

$$Y_k^i(h) = X_k^i - s\xi_k(x;h), \quad Z_k^i = X_k^i + \sqrt{2s}B^i, \quad i = 1,\ldots N,$$

where $B^i$ is the standard Brownian motion. We want to minimize $\mathrm{MMD}(\hat\rho_Y, \hat\rho_Z)$, the MMD between the empirical distribution $\hat\rho_Y(x) = \sum_{i=1}^N \delta(x - Y_k^i(h))$ and $\hat\rho_Z(x) = \sum_{i=1}^N \delta(x - Z_k^i)$. Here, the kernel $K(x,y)$ for the MMD is the Gaussian kernel with a bandwidth of 1. So we optimize over $h$ to minimize $\mathrm{MMD}(\hat\rho_Y, \hat\rho_Z)$, using the bandwidth $h_{k-1}$ from the last iteration as the initialization. For simplicity we denote $\mathrm{BM}(h_{k-1}, \{X_k^i\}_{i=1}^N, s)$ as the output of the BM method.

**Remark 6** Besides KDE, there are other methods that approximate the $\nabla \log \rho_t(x)$ (compute $\xi_k$) via a kernel function, such as the blob method (Carrillo et al., 2019) and the diffusion map (Taghvaei & Mehta, 2019). The BM method can also select the kernel bandwidth for these methods.

## 5.2 ADAPTIVE RESTART

To enhance the practical performance, we introduce an adaptive restart technique, which shares the same idea of gradient restart in (Odonoghue & Candes, 2015; Wang et al., 2019) under the Euclidean case. Consider

$$\varphi_k = -\sum_{i=1}^N \left\langle V_{k+1}^i, \nabla f(X_k^i) + \xi_k(X_k^i) \right\rangle, \tag{13}$$

which can be viewed as discrete-time approximation of $-g_{\rho_t}^W(\partial_t \rho_t, G^W(\rho_t)^{-1}\frac{\delta E}{\delta \rho_t}) = -\partial_t E(\rho_t)$. If $\varphi_k < 0$, then we restart the algorithm with initial values $X_0^i = X_k^i$ and $V_0^i = 0$. This essentially keeps $\partial_t E(\rho_t)$ negative along the trajectory. The numerical results show that the adaptive restart accelerates and stabilizes the discrete-time algorithm. The overall algorithm is summarized below.

---

**Algorithm 1** Particle implementation of Wasserstein AIG flow

---

**Require:** initial positions $\{X_0^i\}_{i=1}^N$, step size $\tau$, number of iteration $L$.
1: Set $k = 0$, $V_0^i = 0$, $i = 1,\ldots N$. Set the bandwidth $h_0$ by MED (12).
2: **for** $l = 1, 2, \ldots L$ **do**
3:     Compute $h_l$ based on BM: $h_l = \mathrm{BM}(h_{l-1}, \{X_k^i\}_{i=1}^N, \sqrt{\tau})$.
4:     Calculate $\xi_k(X_k^i)$ by (10) or by (11) with the bandwidth $h_l$.
5:     Set $\alpha_k$ based on whether $E(\rho)$ is $\mathrm{Hess}(\beta)$ or $\mathrm{Hess}(0)$. For $i = 1, 2, \ldots N$, update
    $V_{k+1}^i = \alpha_k V_k^i - \sqrt{\tau}(\nabla f(X_k^i) + \xi_k(X_k^i))$,    $X_{k+1}^i = X_k^i + \sqrt{\tau}V_{k+1}^i$.
6:     **if** RESTART **then**
7:         Compute $\varphi_k = -\sum_{i=1}^N \left\langle V_{k+1}^i, \nabla f(X_k^i) + \xi_k(X_k^i) \right\rangle$.
8:         If $\varphi_k < 0$, set $X_0^i = X_k^i$ and $V_0^i = 0$ and $k = 0$; otherwise set $k = k + 1$.
9:     **else**
10:        Set $k = k + 1$.
11:     **end if**
12: **end for**

---

## 6 NUMERICAL EXPERIMENTS

In this section, we present several numerical experiments to demonstrate the validity of the BM method, the acceleration effect of the Wasserstein AIG flow, and the strength of the adaptive restart technique. Implementation details can be found in Appendix D.

### 6.1 TOY EXAMPLE

We first investigate the validity of the BM method in selecting the bandwidth. The target density $\rho^*$ is a toy bimodal distribution (Rezende & Mohamed, 2015). We compare two types of particle implementations of the Wasserstein gradient flow over the KL divergence:

$$X_{k+1}^i = X_k^i - \tau\nabla f(X_k^i) + \sqrt{2\tau}B_k^i, \quad X_{k+1}^i = X_k^i - \tau(\nabla f(X_k^i) + \xi_k(X_k^i)).$$

Here $B_k^i \sim \mathcal{N}(0,1)$ is the standard Brownian motion and $\xi_k$ takes the form (11). The first method is known as the MCMC method and the second method is called the ParVI method. For the second method, the bandwidth $h$ is selected by MED/HE/BM respectively. Figure 1 shows the distribution of 200 samples based on different methods. Samples from MCMC match the target distribution in a stochastic way; samples from MED collapse; samples from HE align tidily around the contour lines; samples from BM arrange neatly and are closer to samples from MCMC. This indicates that the BM method makes the particle system behave similar to MCMC, though in a deterministic way.

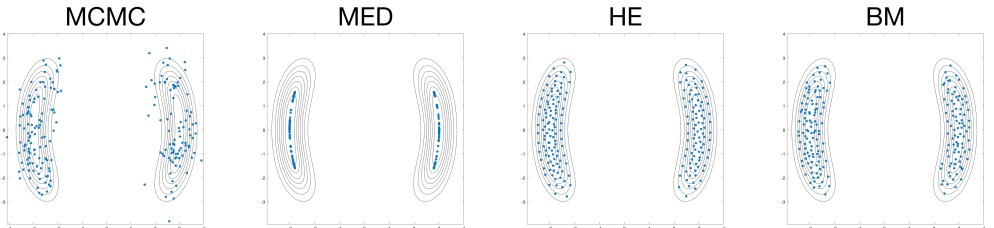

Figure 1: The effect of the BM method. Samples are plotted as blue dots. Left to right: MCMC, MED, HE and BM. All methods are run for 200 iterations with the same initialization.

## 6.2 GAUSSIAN MEASURES

Next, we explore the effectiveness of (W-AIG) flow compared to the Wasserstein gradient flow and demonstrate the strength of the adaptive restart. The target density $\rho^*$ is a Gaussian distribution with zero mean on $R^{100}$, the covariance matrix of $\rho^*$ is $\Sigma^*$ and $W^* = (\Sigma^*)^{-1}$. Let $L$ and $\beta$ be the largest/smallest eigenvalue of $W^*$. $E(\rho)$ satisfies Hess($\beta$) and the step size is $\tau = 1/(4L)$. The condition number of $W^*$ is defined as $\kappa = L/\beta$. The large $L$ indicates $\Sigma^*$ is close to be singular.

We first demonstrate the effectiveness of (W-AIG-G) in the ODE level. Detailed discretization is left in Appendix D.2. The initial value is set to be $\Sigma_0 = I$. For now, WGF denotes the discretization of the Wasserstein gradient flow; AIG-(r)(s) denotes the discretization of the Wasserstein AIG flow. For letters in the parentheses, 'r' denotes using the adaptive restart and 's' denotes utilizing $\beta$. Figure 2 presents the convergence of the KL divergence on two target distributions with small/large $L$. We observe that AIG converges faster compared to WGF, which verifies Theorem 2. The adaptive restart also accelerates the algorithm.

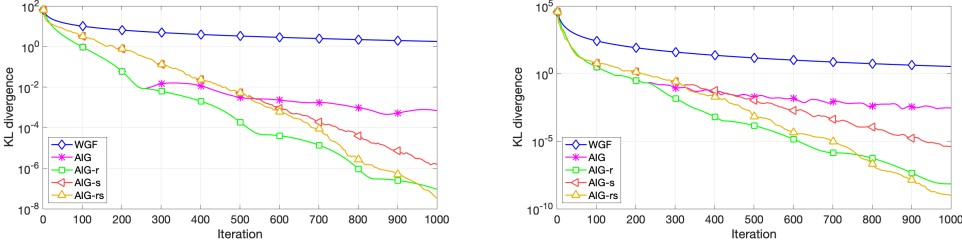

Figure 2: The acceleration effect of W-AIG flow and the strength of adaptive restart (ODE level). The target density is a Gaussian distribution with zero mean on $R^{100}$. Left: $L = 1, \kappa \approx 3.8 \times 10^3$. Right: $\beta = 1, \kappa \approx 4.0 \times 10^3$.

Then, we demonstrate the results in the particle level. The setting of $\rho^*$ is same as the previous experiment. The initial distribution of samples follows $\mathcal{N}(0, I)$ and the number of samples is $N = 600$. For a particle system $\{X_k^i\}_{i=1}^N$, we record the KL divergence $E(\hat{\Sigma}_k)$ (6) using the empirical covariance matrix $\hat{\Sigma}_k$. The left part of Figure 3 (small $L$) is almost identical to Figure 2, which verifies the acceleration effect of AIG flows. It also indicates that the adaptive restart helps to accelerate the convergence. From the right part of Figure 3 (large $L$), AIG and AIG-s diverge because of the ill target distribution, and the adaptive restart solves this problem.

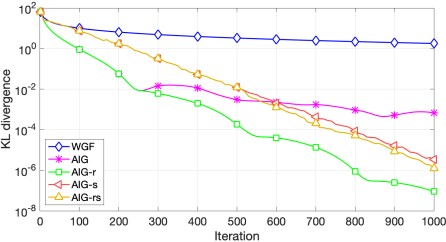 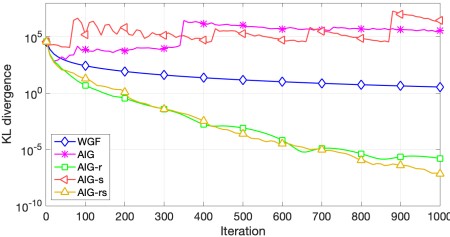

Figure 3: The acceleration effect of W-AIG flow and the strength of adaptive restart (particle level). The setting of target densities is identical to the ones in Figure 2.

## 6.3 BAYESIAN LOGISTIC REGRESSION

We perform the standard Bayesian logistic regression experiment on the Covertype dataset, following the same settings as Liu & Wang (2016). We compare our methods with MCMC, SVGD (Liu & Wang, 2016), WNAG (Liu et al., 2018) and WNes (Liu et al., 2019). We select the bandwidth using either the MED method or the proposed BM method. Figure 4 indicates that the BM method accelerates and stabilizes the performance of WGF and AIG. The performance of MCMC and WGF are similar and they achieve the best log-likelihood. In test accuracy, AIG-r converges faster than other methods and is more stable. The adaptive restart improves the overall performance of AIG.

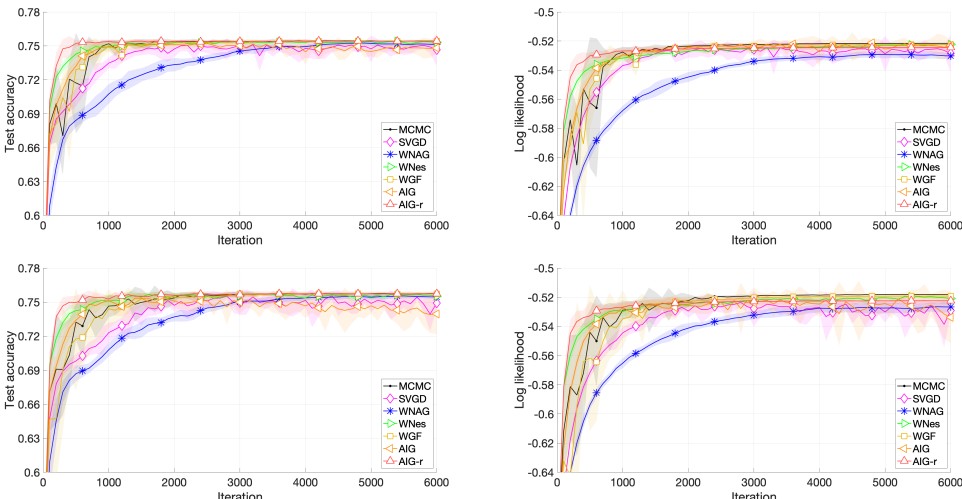

Figure 4: Comparison of different methods on Bayesian logistic regression, averaged over 10 independent trials. The shaded areas show the variance over 10 trials. Top: BM; Bottom: MED. Left: Test accuracy; Right: Test log-likelihood.

## 7 CONCLUSION

In summary, we propose the framework of AIG flows by damping Hamiltonian flows with respect to certain information metrics in the probability space. AIG flows have been carefully studied in Gaussian families. Theoretically, we establish the convergence rate of W-AIG flows. Numerically, we propose the discrete-time algorithm and the adaptive restart technique to overcome the numerical stiffness of W-AIG flows. We introduce a novel kernel selection method by learning from Brownian-motion samples. The numerical experiments verify the acceleration effect of AIG flows and the strength of the adaptive restart. In future works, we intend to systematically explain the stiffness of AIG flows and the effect of the adaptive restart. We shall apply our results to general information metrics, especially for the Fisher-Rao metric. We expect to study the related sampling efficient optimization methods and discrete-time algorithms on general probability models.

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

# A   THE PROOFS AND DERIVATIONS IN SECTION 2

In this section, we provide the derivation of the Euler-Lagrange equation, proofs in Section 2 and the Euler-Lagrange formulation of (AIG).

## A.1   DERIVATION OF THE EULER-LAGRANGE EQUATION

We derive the Euler-Lagrange equation (1) in this subsection. For a fixed $T > 0$ and two given densities $\rho^0, \rho^T$, consider the variational problem

$$I(\rho_t) = \inf_{\rho_t} \left\{ \int_0^T \mathcal{L}(\rho_t, \partial_t \rho_t) dt \,\middle|\, \rho_0 = \rho^0, \rho_T = \rho^T \right\}.$$

Let $h_t \in \mathcal{F}(\Omega)$ be the smooth perturbation function that satisfies $\int h_t dx = 0, t \in [0,T]$ and $h_0 = h_T \equiv 0$. Denote $\rho_t^\epsilon = \rho_t + \epsilon h_t$. Note that

$$I(\rho_t^\epsilon) = \int_0^T \mathcal{L}(\rho_t, \partial_t \rho_t) dt + \epsilon \int_0^T \int \left( \frac{\delta \mathcal{L}}{\delta \rho_t} h_t + \frac{\delta \mathcal{L}}{\delta(\partial_t \rho_t)} \partial_t h_t \right) dx dt + o(\epsilon).$$

From $\frac{dI(\rho_t^\epsilon)}{d\epsilon}\Big|_{\epsilon=0} = 0$, it follows that

$$\int_0^T \int \left( \frac{\delta \mathcal{L}}{\delta \rho_t} h_t + \frac{\delta \mathcal{L}}{\delta(\partial_t \rho_t)} \partial_t h_t \right) dx dt = 0.$$

Note that $h_0 = h_T \equiv 0$. Perform integration by parts w.r.t. $t$ yields

$$\int_0^T \int \left( \frac{\delta \mathcal{L}}{\delta \rho_t} - \partial_t \frac{\delta \mathcal{L}}{\delta(\partial_t \rho_t)} \right) h_t dx dt = 0.$$

Because $\int h_t dx = 0$, (1) holds with a spatially constant function $C(t)$.

## A.2   THE PROOF OF PROPOSITION 1 IN SECTION 2

In this subsection, we derive the Hamiltonian flow in the probability space. First, we give a useful identity. Given a metric tensor $G(\rho) : T_\rho \mathcal{P}(\Omega) \to T_\rho^* \mathcal{P}(\Omega)$, we have

$$\int \sigma_1 G(\rho) \sigma_2 dx = \int G(\rho) \sigma_1 \sigma_2 dx = \int \Phi_1 G(\rho)^{-1} \Phi_2 dx = \int G(\rho)^{-1} \Phi_1 \Phi_2 dx. \tag{14}$$

Here $\Phi_1 = G(\rho)^{-1} \sigma_1$ and $\Phi_2 = G(\rho)^{-1} \sigma_2$. We then check that

$$\frac{\delta}{\delta \rho_t} \left( \int \partial_t \rho_t G(\rho_t) \partial_t \rho_t dx \right) = -\frac{\delta}{\delta \rho_t} \left( \int \Phi_t G(\rho_t)^{-1} \Phi_t dx \right). \tag{15}$$

Let $\tilde{\rho}_t = \rho_t + \epsilon h$, where $h \in T_{\rho_t} \mathcal{P}(\Omega)$. For all $\sigma \in T_{\rho_t} \mathcal{P}$, it follows

$$G(\rho_t + \epsilon h)^{-1} G(\rho_t + \epsilon h) \sigma = \sigma.$$

The first-order derivative w.r.t. $\epsilon$ of the left hand side shall be 0, i.e.,

$$\left( \frac{\partial G(\rho_t)^{-1}}{\partial \rho_t} \cdot h \right) G(\rho_t) \sigma + G(\rho_t)^{-1} \left( \frac{\partial G(\rho_t)}{\partial \rho_t} \cdot h \right) \sigma = 0.$$

Because $\partial_t \rho_t = G(\rho)^{-1} \Phi_t$, applying (14) yields

$$\int \partial_t \rho_t \left( \frac{\partial G(\rho_t)}{\partial \rho_t} \cdot h \right) \partial_t \rho_t dx = \int \Phi_t G(\rho_t)^{-1} \left( \frac{\partial G(\rho_t)}{\partial \rho_t} \cdot h \right) \partial_t \rho_t dx$$
$$= -\int \Phi_t \left( \frac{\partial G(\rho_t)^{-1}}{\partial \rho_t} \cdot h \right) G(\rho_t) \partial_t \rho_t dx = -\int \Phi_t \left( \frac{\partial G(\rho_t)^{-1}}{\partial \rho_t} \cdot h \right) \Phi_t dx. \tag{16}$$

Based on basic calculations, we can compute that

$$\int \partial_t \rho_t G(\tilde{\rho}_t) \partial_t \rho_t dx - \int \partial_t \rho_t G(\rho_t) \partial_t \rho_t dx = \epsilon \int \partial_t \rho_t \left( \frac{\partial G(\rho_t)}{\partial \rho_t} \cdot h \right) \partial_t \rho_t dx + o(\epsilon), \quad (17)$$

$$- \int \Phi_t G(\tilde{\rho}_t)^{-1} \Phi_t dx + \int \Phi_t G(\rho_t)^{-1} \Phi_t dx = -\epsilon \int \Phi_t \left( \frac{\partial G(\rho_t)^{-1}}{\partial \rho_t} \cdot h \right) \Phi_t dx + o(\epsilon). \quad (18)$$

Combining (16), (17) and (18) yields (15). Hence, the Euler-Lagrange equation (1) is equivalent to

$$\partial_t \Phi_t = \frac{1}{2} \frac{\delta}{\delta \rho_t} \left( \int \partial_t \rho_t G(\rho_t) \partial_t \rho_t dx \right) - \frac{\delta E}{\delta \rho_t} = -\frac{1}{2} \frac{\delta}{\delta \rho_t} \left( \int \Phi_t G(\rho_t)^{-1} \Phi_t dx \right) - \frac{\delta E}{\delta \rho_t}.$$

This equation combining with $\partial_t \rho_t = G(\rho)^{-1} \Phi_t$ recovers the Hamiltonian flow (2). In short, the Euler-Lagrange equation (1) is from the primal coordinates $(\rho_t, \partial_t \rho_t)$ and the Hamiltonian flow (2) is from the dual coordinates $(\rho_t, \Phi_t)$. Similar interpretations can be found in (Chow et al., 2019).

### A.3 The Euler-Lagrangian formulation of AIG flows

We can formulate (AIG) as a second-order equation of $\rho_t$,

$$\frac{D^2}{Dt^2} \rho_t + \alpha_t \partial_t \rho_t + G(\rho_t)^{-1} \frac{\delta E}{\delta \rho_t} = 0.$$

Here $D^2/Dt^2$ is the covariant derivative in metric $G$. We can also explicitly write $\frac{D^2}{Dt^2} \rho_t$ as follows.

$$\frac{D^2}{Dt^2} \rho_t = \partial_{tt} \rho_t - (\partial_t G(\rho_t)^{-1}) \partial_t \rho_t + \frac{1}{2} G(\rho_t)^{-1} \frac{\delta}{\delta \rho_t} \left( \int \partial_t \rho_t G(\rho_t) \partial_t \rho_t dx \right).$$

## B The proofs in Section 3

In this section, we present proofs of propositions and theorems in Section 3.

### B.1 The proof of Proposition 2 in Section 3

We start with an identity. For a twice differentiable $\Phi(x)$, we have

$$\frac{1}{2} \nabla \|\nabla \Phi\|^2 = \nabla^2 \Phi \nabla \Phi = (\nabla \Phi \cdot \nabla) \nabla \Phi. \quad (19)$$

From (W-AIG), it follows that

$$\partial_t \rho_t + \nabla \cdot (\rho_t \nabla \Phi_t) = 0. \quad (20)$$

This is the continuity equation of $\rho_t$. Hence, on the particle level, $X_t$ shall follows

$$dX_t = \nabla \Phi_t(X_t) dt.$$

Let $V_t = \nabla \Phi_t(X_t)$. Then, by the conservation of momentum and (W-AIG), we have

$$dV_t = (\partial_t + \nabla \Phi_t(X_t) \cdot \nabla) \nabla \Phi_t(X_t) dt$$
$$= \left( -\alpha_t \nabla \Phi_t(X_t) - \frac{1}{2} \nabla \|\nabla \Phi\|^2 - \nabla \frac{\delta E}{\delta \rho_t} \right) dt + (\nabla \Phi \cdot \nabla) \nabla \Phi dt$$
$$= -\alpha_t \nabla \Phi_t(X_t) dt - \nabla \frac{\delta E}{\delta \rho_t}(X_t) dt = -\alpha_t V_t dt - \nabla \frac{\delta E}{\delta \rho_t}(X_t) dt.$$

### B.2 The proof of Proposition 3 in Subsection 3.1

In this subsection, we derive the Hamiltonian flow in Gaussian. For $A \in \mathbb{S}^n$, we define the linear operator $M_A : \mathbb{S}^n \to \mathbb{S}^n$ by

$$M_A B = AB + BA, \quad B \in \mathbb{S}^n.$$

It is easy to verify that if $A \in \mathbb{P}^n$, then $M_A^{-1}$ is well-defined. For a flow $\Sigma_t \in \mathbb{P}^n, t \geq 0$, we define the Lagrangian $L(\Sigma_t, \dot{\Sigma}_t) = \frac{1}{2} g_{\Sigma_t}(\dot{\Sigma}_t, \dot{\Sigma}_t) - E(\Sigma_t)$. The corresponding Euler-Lagrange equation writes

$$\frac{d}{dt} \frac{dL}{d\dot{\Sigma}_t} = \frac{dL}{d\Sigma}. \tag{21}$$

Let $S_t = M_{\Sigma_t}^{-1} \dot{\Sigma}_t$, i.e., $\dot{\Sigma}_t = S_t \Sigma_t + \Sigma_t S_t$. Then, it follows

$$g_{\Sigma_t}(\dot{\Sigma}_t, \dot{\Sigma}_t) = \text{tr}(S_t \Sigma_t S_t) = \frac{1}{2} \text{tr}((S_t \Sigma_t + \Sigma_t S_t) S_t) = \frac{1}{2} \text{tr}(\dot{\Sigma}_t S_t) = \frac{1}{2} \text{tr}(\dot{\Sigma}_t M_{\Sigma_t}^{-1} \dot{\Sigma}_t).$$

This leads to $\frac{dL}{d\dot{\Sigma}_t} = M_{\Sigma_t}^{-1} \dot{\Sigma}_t = S_t$. For simplicity, we denote $g = g_{\Sigma_t}(\dot{\Sigma}_t, \dot{\Sigma}_t)$. First, we show that

$$\frac{dg}{d\Sigma_t} = -S_t^2.$$

Because $S_t = M_{\Sigma_t}^{-1} \dot{\Sigma}_t$. Given $\dot{\Sigma}_t$, $S_t$ can be viewed as a continuous function of $\Sigma_t$. For any $A \in \mathbb{S}^n$, define $l_A = \text{tr}((\Sigma_t S_t + S_t \Sigma_t) A)$.

$$0 = \frac{dl_A}{d\Sigma_t} = \frac{\partial S_t}{\partial \Sigma_t} \frac{\partial L_A}{\partial S_t} + \frac{\partial l_A}{\partial \Sigma_t} = \frac{\partial S_t}{\partial \Sigma_t}(A\Sigma_t + \Sigma_t A) + (AS_t + S_t A).$$

Here we view $\partial S_T / \partial \Sigma_t$ as a linear operator on $S^n$. Let $B = A\Sigma_t + \Sigma_t A$, then $A = M_{\Sigma_t}^{-1} B$. $\frac{\partial S_t}{\partial \Sigma_t} B + M_{S_t} M_{\Sigma_t}^{-1} B = 0$ holds for all $B \in S^n$. Therefore, we have $\frac{\partial S_t}{\partial \Sigma_t} = -M_{S_t} M_{\Sigma_t}^{-1}$. Hence,

$$\frac{dg}{d\Sigma_t} = \frac{\partial S_t}{\partial \Sigma_t} \frac{\partial g}{\partial S_t} + \frac{\partial g}{\partial \Sigma_t} = -M_{S_t} M_{\Sigma_t}^{-1}(S_t \Sigma_t + \Sigma_t S_t) + S_t^2 = -M_{S_t} S_t + S_t^2 = -S_t^2.$$

As a result, the Euler-Lagrange equation (21) is equivalent to

$$\frac{\dot{S}}{2} = \frac{dL}{d\Sigma_t} = -\frac{S_t^2}{2} - \nabla E(\Sigma_t). \tag{22}$$

Combining (22) with $\dot{\Sigma}_t = S_t \Sigma_t + \Sigma_t S_t$ renders the Hamiltonian flow (5).

### B.3 The proof of Theorem 1 in Subsection 3.1

We first show that $\Sigma_t$ stays in $\mathbb{P}^n$. Suppose that $\Sigma_t \in \mathbb{P}_n$ for $0 \leq t \leq T$. Define $H_t = H(\Sigma_t, S_t) = \text{tr}(S_t \Sigma_t S_t)/2 + E(\Sigma_t)$. We observe that (W-AIG-G) is equivalent to

$$\dot{\Sigma}_t = 2\frac{\partial H_t}{\partial S_t}, \quad \dot{S}_t = -\alpha_t S_t - 2\frac{\partial H_t}{\partial \Sigma_t}. \tag{23}$$

We show that $H_t$ is decreasing with respect to $t$.

$$\frac{dH_t}{dt} = \text{tr}\left(\frac{\partial H_t}{\partial S_t} \dot{S}_t + \frac{\partial H_t}{\partial \Sigma_t} \dot{\Sigma}_t\right) = \text{tr}\left(\frac{\partial H_t}{\partial S_t}\left(-\alpha_t S_t - 2\frac{\partial H_t}{\partial \Sigma_t}\right) + 2\frac{\partial H_t}{\partial \Sigma_t} \frac{\partial H_t}{\partial S_t}\right)$$

$$= -\alpha_t \text{tr}\left(S_t \frac{\partial H_t}{\partial S_t}\right) = -\frac{\alpha_t}{2} \text{tr}(S_t(\Sigma_t S_t + S_t \Sigma_t)) = -\alpha_t \text{tr}(S_t \Sigma_t S_t) \leq 0.$$

For simplicity, we denote $W^* = (\Sigma^*)^{-1}$. Let $\lambda_t$ be the smallest eigenvalue of $\Sigma_t$. Then, $\log \det(\Sigma_t W^*) \geq \log \det(\Sigma_t) = n \log \lambda_t$. Therefore,

$$-\frac{n}{2}(\log \lambda_t + 1) \leq -\frac{1}{2}[\log \det(\Sigma_t W^*) + n] \leq E(\Sigma_t) \leq H(t) \leq H(0),$$

which indicates that

$$\lambda_t \geq \exp\left(-\frac{2}{n}H(0) - 1\right). \tag{24}$$

This means that as long as $\Sigma_t \in \mathbb{P}_n$, the smallest eigenvalue of $\Sigma_t$ has a positive lower bounded. If there exists $T > 0$ such that $\Sigma_T \notin \mathbb{P}_n$. Because $\Sigma_t$ is continuous with respect to $t$, there exists $T_1 < T$, such that $\Sigma_t \in P_n, 0 \leq t \leq T_1$ and $\lambda_{T_1} < \exp(-2H(0)/n - 1)$, which violates (24).

We then reveal the relationship between (W-AIG) in $\mathcal{P}(\mathbb{R}^n)$ and $\mathbb{P}^n$. We can compute that

$$\frac{\partial}{\partial t}\det(\Sigma_t) = \det(\Sigma_t)\operatorname{tr}(\Sigma_t^{-1}\dot{\Sigma}_t), \quad \frac{\partial}{\partial t}\Sigma_t^{-1} = -\Sigma_t^{-1}\dot{\Sigma}_t\Sigma_t^{-1}.$$

Combining with $\dot{\Sigma}_t = \Sigma_t S_t + S_t \Sigma_t$, we obtain

$$\operatorname{tr}(\Sigma_t^{-1}\dot{\Sigma}_t) = \operatorname{tr}(S_t + \Sigma_t^{-1}S_t\Sigma_t) = 2\operatorname{tr}(S_t),$$
$$\operatorname{tr}(x\Sigma_t^{-1}\dot{\Sigma}_t\Sigma_t^{-1}x) = \operatorname{tr}(x^T\Sigma_t^{-1}S_t x + x^T S_t \Sigma_t^{-1}x) = 2\operatorname{tr}(S_t\Sigma_t^{-1}xx^T).$$

Therefore, it follows

$$\partial_t \rho_t(x) = \frac{\partial}{\partial t}\left(\frac{1}{\sqrt{\det(\Sigma_t)}}\right)\sqrt{\det(\Sigma_t)}\rho_t(x) + \frac{1}{2}\operatorname{tr}(x^T\Sigma_t^{-1}\dot{\Sigma}_t\Sigma_t^{-1}x)\rho_t(x)$$

$$= -\frac{1}{2}\operatorname{tr}(\Sigma_t^{-1}\dot{\Sigma}_t)\rho_t(x) + \operatorname{tr}(S_t\Sigma_t^{-1}xx^T)\rho_t(x) = -\operatorname{tr}(S_t(I - \Sigma_t^{-1}xx^T))\rho_t(x).$$

Note that $\nabla\Phi_t(x) = S_t x$. Hence,

$$-\nabla\cdot(\rho_t\nabla\Phi_t) = -\sum_{i=1}^{n}\partial_i(\rho_t(x)S_t x)_i = -\sum_{i=1}^{n}[\rho_t(x)\partial_i(S_t x)_i + (S_t x)_i\partial_i\rho_t(x)]$$

$$= -\rho_t(x)\left[\operatorname{tr}(S_t) + (S_t x)^T(-\Sigma_t^{-1}x)\right] = -\rho_t(x)\operatorname{tr}(S_t(I - \Sigma_t^{-1}xx^T)) = \partial_t\rho_t(x).$$

The first equation of (W-AIG) holds. Because $\partial_t\Phi_t(x) = x^T\dot{S}_t x/2 + \dot{C}(t)$,

$$\partial_t\Phi_t(x) + \alpha_t\Phi_t(x) + \frac{1}{2}\|\nabla\Phi_t(x)\|^2 = \frac{1}{2}x^T\dot{S}_t x + \frac{\alpha_t}{2}x^T S_t x + \frac{1}{2}x^T S_t^2 x + \dot{C}(t)$$

$$= -x^T\nabla_{\Sigma_t}E(\Sigma_t)x + \dot{C}(t) = \frac{1}{2}x^T(\Sigma_t^{-1} - W^*)x + \dot{C}(t).$$

Note that $\rho^*$ is the Gaussian density with the covariance matrix $\Sigma^*$. Because $\dot{C}(t) = \frac{1}{2}\log\det(\Sigma_t W^*) - 1$, we can compute

$$\frac{\delta E}{\delta\rho_t} = \log\rho_t(x) - \log\rho^*(x) + 1 = -\frac{1}{2}x^T(\Sigma_t^{-1} - W^*)x - \frac{1}{2}\log\det(\Sigma_t W^*) + 1$$

$$= -\frac{1}{2}x^T(\Sigma_t^{-1} - W^*)x - \dot{C}(t) = -(\partial_t\Phi_t(x) + \alpha_t\Phi_t(x) + \frac{1}{2}\|\nabla\Phi_t(x)\|^2).$$

Therefore, the second equation of (W-AIG) holds. Because $\Sigma_0 = \Sigma^0$, $S_0 = 0$ and $C(0) = 0$, we have $\rho_0 = \rho^0$ and $\Phi_0 = 0$. This completes the proof.

## C  THE PROOFS IN SECTION 4

In this section, we briefly review some geometric properties of the probability space as a Riemannian manifold and present proofs of propositions in Section 4.

### C.1  A BRIEF REVIEW ON THE GEOMETRIC PROPERTIES OF THE PROBABILITY SPACE

Suppose that we have a metric $g_\rho$ in the probability space $\mathcal{P}(\Omega)$. Given two probability densities $\rho^0, \rho^1 \in \mathcal{P}(\Omega)$, we define the distance as follows

$$d(\rho^0, \rho^1) = \left(\inf_{\gamma(s)}\left\{\int_0^1 g_{\gamma(s)}(\dot{\gamma}(s), \dot{\gamma}(s))ds : \gamma(0) = \rho^0, \gamma(1) = \rho^1\right\}\right)^{1/2}.$$

The minimizer $\gamma(s)$ of the above problem is defined as the geodesic curve connecting $\rho^0$ and $\rho^1$. An exponential map at $\rho^0 \in \mathcal{P}(\Omega)$ is a mapping from the tangent space $T_{\rho^0}\mathcal{P}(\Omega)$ to $\mathcal{P}(\Omega)$. It requires that $\sigma \in T_{\rho^0}\mathcal{P}(\Omega)$ is mapped to a point $\rho^1 \in \mathcal{P}(\Omega)$ such that there exists a geodesic curve $\gamma(s)$ satisfying $\gamma(0) = \rho^0$, $\dot{\gamma}(0) = \sigma$ and $\gamma(1) = \rho^1$.

## C.2  THE PROOF OF PROPOSITION 4 IN SECTION 4

In this subsection, we characterize the inverse of the exponential map in the probability space with the Wasserstein metric. Let $T_t^s = (sT_t + (1-s)\,\mathrm{Id})^{-1}, s \in [0,1]$. Then, based on the theory of optimal transport (Villani, 2003), we can write the explicit formula of the geodesic curve $\gamma(s)$ by

$$\gamma(s) = T_t^s \# \rho_t = \det(\nabla T_t^s) \rho_t \circ T_t^s.$$

Through basic calculations, we can compute that

$$\frac{d}{ds} T_t^s \Big|_{s=0} = -\frac{d}{ds}(sT_t + (1-s)\,\mathrm{Id})\Big|_{s=0} = \mathrm{Id} - T_t.$$

$$\frac{d}{ds} \det(\nabla T_t^s)\Big|_{s=0} = \frac{d}{ds}\det(I + s(I - DT_t) + o(s))\Big|_{s=0} = \mathrm{tr}(I - DT_t).$$

Therefore,

$$\frac{d}{ds}\gamma(s)\Big|_{s=0}(x) = \mathrm{tr}(I - \nabla T_t)\rho_t(x) + \langle \nabla \rho_t(x), x - \varphi_t(x)\rangle$$

$$= \nabla \cdot (x - T_t(x))\rho_t(x) + \langle \nabla \rho_t(x), x - T_t(x)\rangle = -\nabla \cdot (\rho_t(x)(T_t(x) - x)),$$

which completes the proof.

## C.3  THE PROOF OF PROPOSITION 5 AND 6 IN SECTION 4

The main goal of this subsection is to prove the Lyapunov function $\mathcal{E}(t)$ is non-increasing.

**Preparations.** We first give a better characterization of the optimal transport plan $T_t$. We can write $T_t = \nabla \Psi_t$, where $\Psi_t$ is a strictly convex function, see (Villani, 2003). This indicates that $\nabla T_t$ is symmetric. We then introduce the following proposition.

**Proposition 7** *Suppose that $E(\rho)$ satisfies Hess($\beta$) for $\beta \geq 0$. Let $T_t(x)$ be the optimal transport plan from $\rho_t$ to $\rho^*$, then*

$$E(\rho^*) \geq E(\rho_t) + \int \left\langle T_t(x) - x, \nabla \frac{\delta E}{\delta \rho_t}\right\rangle \rho dx + \frac{\beta}{2} \int \|T_t(x) - x\|^2 \rho_t dx.$$

This is a direct result of $\beta$-geodesic convexity of $E(\rho)$ based on Proposition 4.

Next, let us denote $u_t = \partial_t (T_t)^{-1} \circ T_t$. We show that $u_t$ satisfies

$$\nabla \cdot (\rho_t(u_t - \nabla \Phi_t)) = 0. \tag{25}$$

Because $(T_t)^{-1} \# \rho^* = \rho_t$, let $u_t = \partial_t (T_t)^{-1} \circ T_t$ and $X_t = (T_t)^{-1} X_0$, where $X_0 \sim \rho^*$. This yields $\frac{d}{dt} X_t = u_t(X_t)$. The distribution of $X_t$ follows $\rho_t$. By the Euler's equation, $\rho_t$ shall follows

$$\partial_t \rho_t + \nabla \cdot (\rho_t u_t) = 0.$$

Combining this with the continuity equation (20) yields (25).

Then, we formulate $\partial_t T_t(x)$ with $u_t$. By the Taylor expansion,

$$T_{t+s}(x) = T_t(x) + s\partial_t T_t(x) + o(s).$$

Let $y = (T_t)^{-1}x$. it follows

$$(T_{t+s})^{-1}(x) = (T_t)^{-1}(x) + su_t((T_t)^{-1}(x)) + o(s) = y + su_t(y) + o(s).$$

Therefore,

$$0 = T_{t+s}((T_{t+s})^{-1}(x)) - x = T_{t+s}(y + su_t(y) + o(s)) - x$$

$$= T_t(y + su_t(y)) + s\partial_t T_t(y + su_t(y)) - x + o(s)$$

$$= T_t(y) + s\nabla T_t(y)u_t(y) + s\partial_t T_t(y) - x + o(s)$$

$$= s\left[\nabla T_t(y)u_t(y) + \partial_t T_t(y)\right] + o(s).$$

We shall have $\nabla T_t(y)u_t(y) + \partial_t T_t(y) = 0$. Replacing $y$ by $x$ gives

$$\partial_t T_t(x) = -\nabla T_t(x)u_t(x). \tag{26}$$

The following lemma illustrates two important properties of $u_t$ and $\partial_t T_t$.

**Lemma 1** *For $u_t$ satisfying (25), we have*

$$\int \langle \nabla \Phi_t - u_t, \nabla T_t \nabla \Phi_t \rangle \rho_t dx \geq 0, \quad \int \langle \nabla \Phi_t - u_t, \nabla T_t(x)(T_t(x) - x) \rangle \rho_t = 0.$$

PROOF We first notice that $u_t - \nabla \Phi_t$ is divergence-free in term of $\rho_t$. From $-\nabla T_t u_t = \partial_t T_t = \nabla \partial_t \Psi_t$, we observe that $-\nabla T_t u_t$ is the gradient of $\partial_t \Psi_t$. Therefore,

$$\int \langle \nabla \Phi_t - u_t, \nabla T_t u_t \rangle \rho_t = -\int \langle \partial_t \Psi_t, \nabla \cdot (\rho_t(\nabla \Phi_t - u_t)) \rangle = 0.$$

Based on our previous characterization on the optimal transport plan $T_t$, $\nabla T_t = \nabla^2 \Psi_t$ is symmetric positive definite. This yields that

$$\int \langle \nabla \Phi_t - u_t, \nabla T_t \nabla \Phi_t \rangle \rho_t dx = \int \langle \nabla \Phi_t - u_t, \nabla T_t \nabla \Phi_t \rangle \rho_t dx - \int \langle \nabla \Phi_t - u_t, \nabla T_t u_t \rangle \rho_t$$

$$= \int \langle \nabla \Phi_t - u_t, \nabla T_t(\nabla \Phi_t - u_t) \rangle \rho_t dx \geq 0.$$

The last inequality utilizes that $\nabla T_t$ is positie definite and $\rho_t$ is non-negative. Then, we prove the equality in Lemma 1. Because $\nabla T_t(x)(T_t(x) - x) = \frac{1}{2}\nabla(\|T_t(x) - x\|^2 + T_t(x) - \|x\|^2)$ is a gradient. Similarly, it follows

$$\int \langle \nabla \Phi_t - u_t, \nabla T_t(x)(T_t(x) - x) \rangle \rho_t = 0.$$

Lemma 1 and the relationship (26) gives

$$-\int \langle \partial_t T_t, \nabla \Phi_t \rangle \rho_t dx = \int \langle u_t, \nabla T_t \nabla \Phi_t \rangle \rho_t dx \leq \int \langle \nabla \Phi_t, \nabla T_t \nabla \Phi_t \rangle \rho_t dx, \qquad (27)$$

$$\int \langle \partial_t T_t, T_t(x) - x \rangle \rho_t dx = -\int \langle \nabla \Phi_t, \nabla T_t(x)(T_t(x) - x) \rangle \rho_t dx. \qquad (28)$$

**Proof of Proposition 5.** Based on the definition of the Wasserstein metric, we have

$$\partial_t E(\rho_t) = -\int \frac{\delta E}{\delta \rho_t} \nabla \cdot (\rho_t \nabla \Phi_t) dx.$$

Differentiating $\mathcal{E}(t)$ (7) w.r.t. $t$ renders

$$\dot{\mathcal{E}}(t)e^{-\sqrt{\beta}t} = \beta \int \langle \partial_t T_t, T_t(x) - x \rangle \rho_t dx - \frac{\beta}{2} \int \|T_t(x) - x\|^2 \nabla \cdot (\rho_t \nabla \Phi_t) dx$$

$$- \sqrt{\beta} \int \langle \partial_t T_t, \nabla \Phi_t \rangle \rho_t dx - \sqrt{\beta} \int \langle T_t(x) - x, \partial_t \nabla \Phi_t \rangle \rho_t dx$$

$$+ \sqrt{\beta} \int \langle T_t(x) - x, \nabla \Phi_t \rangle \nabla \cdot (\rho_t \nabla \Phi_t) dx + \int \langle \nabla \Phi_t, \partial_t \nabla \Phi_t \rangle \rho_t dx$$

$$- \frac{1}{2} \int \|\nabla \Phi_t\|^2 \nabla \cdot (\rho_t \nabla \Phi_t) - \int \frac{\delta E}{\delta \rho_t} \nabla \cdot (\rho_t \nabla \Phi_t) dx$$

$$+ \frac{\sqrt{\beta}}{2} \int \|\nabla \Phi_t\|^2 \rho_t dx - \beta \int \langle T_t(x) - x, \nabla \Phi_t(x) \rangle \rho_t dx$$

$$+ \frac{\sqrt{\beta^3}}{2} \int \|T_t(x) - x\|^2 \rho_t dx + \sqrt{\beta}(E(\rho_t) - E(\rho^*)). \qquad (29)$$

For the part (29), Proposition 7 renders

$$\frac{\sqrt{\beta^3}}{2} \int \|T_t(x) - x\|^2 \rho_t dx + \sqrt{\beta}E(\rho_t) \leq -\sqrt{\beta} \int \left\langle T_t(x) - x, \nabla \frac{\delta E}{\delta \rho_t} \right\rangle \rho_t dx. \qquad (30)$$

We first compute the terms with the coefficient $\beta^0$ in $\dot{\mathcal{E}}(t)e^{-\sqrt{\beta}t}$. We observe that

$$\int \langle \nabla \Phi_t, \partial_t \Phi_t \rangle \rho_t dx - \frac{1}{2} \int \|\nabla \Phi_t\|^2 \nabla \cdot (\rho_t \nabla \Phi_t) dx - \int \frac{\delta E}{\delta \rho_t} \nabla \cdot (\rho_t \nabla \Phi_t) \rho_t dx$$

$$= \int \left\langle \partial_t \nabla \Phi_t + \frac{1}{2} \nabla \|\nabla \Phi_t\|^2 + \nabla \frac{\delta E}{\delta \rho}, \nabla \Phi_t \right\rangle \rho_t dx = -2\sqrt{\beta} \int \|\nabla \Phi_t\|^2 \rho_t dx, \qquad (31)$$

where the last equality uses (W-AIG) with $\alpha_t = 2\sqrt{\beta}$. Substituting (30) and (31) into the expression of $\dot{\mathcal{E}}(t)e^{-\sqrt{\beta}t}$ yields

$$
\begin{aligned}
\dot{\mathcal{E}}(t)e^{-\sqrt{\beta}t} \leq & \beta \int \langle \partial_t T_t, T_t(x) - x \rangle \rho_t dx - \frac{\beta}{2} \int \|T_t(x) - x\|^2 \nabla \cdot (\rho_t \nabla \Phi_t) dx \\
& - \beta \int \langle T_t(x) - x, \nabla \Phi_t \rangle \rho_t dx - \sqrt{\beta} \int \langle \partial_t T_t, \nabla \Phi_t \rangle \rho_t dx \\
& - \sqrt{\beta} \int \langle T_t(x) - x, \partial_t \nabla \Phi_t \rangle \rho_t dx - \sqrt{\beta} \int \left\langle T_t(x) - x, \nabla \frac{\delta E}{\delta \rho_t} \right\rangle \rho_t dx \\
& + \sqrt{\beta} \int \langle T_t(x) - x, \nabla \Phi_t \rangle \nabla \cdot (\rho_t \nabla \Phi_t) dx - \frac{3\sqrt{\beta}}{2} \int \|\nabla \Phi_t\|^2 \rho_t dx.
\end{aligned} \tag{32}
$$

Then, we deal with the terms with $\nabla \cdot (\rho_t \nabla \Phi_t)$. We have the following two identities

$$
\begin{aligned}
& \int \langle T_t(x) - x, \nabla \Phi_t \rangle \nabla \cdot (\rho_t \nabla \Phi_t) dx = - \int \langle \nabla \langle T_t(x) - x, \nabla \Phi_t \rangle, \nabla \Phi_t \rangle \rho_t dx \\
= & - \int \langle \nabla \Phi_t, \nabla^2 \Phi_t(x)(T_t(x) - x) + (\nabla T_t(x) - I)\nabla \Phi_t \rangle \rho_t dx \\
= & - \frac{1}{2} \int \langle T_t(x) - x, \nabla \|\nabla \Phi_t\|^2 \rangle \rho_t dx - \int \langle \nabla \Phi_t, \nabla T_t \nabla \Phi_t \rangle \rho_t dx + \int \|\nabla \Phi_t\|^2 \rho_t dx.
\end{aligned} \tag{33}
$$

$$
\begin{aligned}
& - \frac{1}{2} \int \|T_t(x) - x\|^2 \nabla \cdot (\rho_t \nabla \Phi_t) dx = \int \langle (\nabla T_t(x) - I)(T_t(x) - x), \nabla \Phi_t \rangle \rho_t dx \\
= & \int \langle T_t(x) - x, \nabla T_t \nabla \Phi_t \rangle \rho_t dx - \int \langle T_t(x) - x, \nabla \Phi_t \rangle \rho_t dx.
\end{aligned} \tag{34}
$$

Hence, we can proceed to compute the terms with the coefficient $\sqrt{\beta}$. (27) and (33) yields

$$
\begin{aligned}
& - \sqrt{\beta} \int \langle \partial_t T_t, \nabla \Phi_t \rangle \rho_t dx - \sqrt{\beta} \int \left\langle T_t(x) - x, \partial_t \nabla \Phi_t + \nabla \frac{\delta E}{\delta \rho_t} \right\rangle \rho_t dx \\
& - \frac{3\sqrt{\beta}}{2} \int \|\nabla \Phi_t\|^2 \rho_t dx + \sqrt{\beta} \int \langle T_t(x) - x, \nabla \Phi_t \rangle \nabla \cdot (\rho_t \nabla \Phi_t) dx \\
= & - \sqrt{\beta} \int \langle \partial_t T_t + \nabla T_t \nabla \Phi_t, \nabla \Phi_t \rangle \rho_t dx - \frac{\sqrt{\beta}}{2} \int \|\nabla \Phi_t\|^2 \rho_t dx \\
& - \sqrt{\beta} \int \left\langle T_t(x) - x, \partial_t \nabla \Phi_t + \nabla \frac{\delta E}{\delta \rho} + \frac{1}{2}\nabla \|\nabla \Phi_t\|^2 \right\rangle \rho_t dx \\
\leq & - \frac{\sqrt{\beta}}{2} \int \|\nabla \Phi_t\|^2 \rho_t dx + 2\beta \int \langle T_t(x) - x, \nabla \Phi_t \rangle \rho_t dx.
\end{aligned} \tag{35}
$$

Substituting (34) and (35) into (32) gives

$$
\begin{aligned}
& \dot{\mathcal{E}}(t)e^{-\sqrt{\beta}t} + \frac{\sqrt{\beta}}{2} \int \|\nabla \Phi_t\|^2 \rho_t dx \\
\leq & \beta \int \langle \partial_t T_t, T_t(x) - x \rangle \rho_t dx - \frac{\beta}{2} \int \|T_t(x) - x\|^2 \nabla \cdot (\rho_t \nabla \Phi_t) dx \\
& - \beta \int \langle T_t(x) - x, \nabla \Phi_t \rangle \rho_t dx + 2\beta \int \langle T_t(x) - x, \nabla \Phi_t \rangle \rho_t dx \\
= & \beta \int \langle \partial_t T_t + \nabla T_t \nabla \Phi_t, T_t(x) - x \rangle \rho_t dx = 0,
\end{aligned}
$$

where the last equality uses (28). In summary, we have

$$
\dot{\mathcal{E}}(t)e^{-\sqrt{\beta}t} \leq -\frac{\sqrt{\beta}}{2} \int \|\nabla \Phi_t\|^2 \rho_t dx \leq 0.
$$

**Proof of Proposition 6.** Differentiating $\mathcal{E}(t)$ (8) w.r.t. $t$, we compute that

$$\dot{\mathcal{E}}(t) = \int \langle \partial_t T_t, T_t(x) - x \rangle \, \rho_t dx - \frac{1}{2} \int \|T_t(x) - x\|^2 \nabla \cdot (\rho_t \nabla \Phi_t) dx$$

$$- \int \left\langle \partial_t T_t, \frac{t}{2} \nabla \Phi_t \right\rangle \rho_t dx - \int \left\langle T_t(x) - x, \frac{1}{2} \nabla \Phi_t + \frac{t}{2} \partial_t \nabla \Phi_t \right\rangle \rho_t dx$$

$$+ \int \left\langle T_t(x) - x, \frac{t}{2} \nabla \Phi_t \right\rangle \nabla \cdot (\rho_t \nabla \Phi_t) dx + \int \left\langle \frac{t}{2} \nabla \Phi_t, \frac{1}{2} \nabla \Phi_t + \frac{t}{2} \partial_t \nabla \Phi_t \right\rangle \rho_t dx$$

$$- \frac{1}{2} \int \left\| \frac{t}{2} \nabla \Phi_t \right\|^2 \nabla \cdot (\rho_t \nabla \Phi_t) dx - \frac{t^2}{4} \int \frac{\delta E}{\delta \rho_t} \nabla \cdot (\rho_t \nabla \Phi_t) dx + \frac{t}{2} (E(\rho_t) - E(\rho^*)). \tag{36}$$

Because $E(\rho)$ is Hess(0), Proposition 7 yields

$$E(\rho_t) = E(\rho_t) - E(\rho^*) \leq - \int \left\langle T_t(x) - x, \nabla \frac{\delta E}{\delta \rho_t} \right\rangle \rho_t dx. \tag{37}$$

Utilizing the inequality (37) and substituting the expressions of terms involving $\partial_t T_t$ and $\nabla \cdot (\rho_t \nabla \Phi_t)$ in (36) with the expressions in (27) (28) and (33) (34), we obtain

$$\dot{\mathcal{E}}(t) \leq - \int \langle \nabla \Phi_t, \nabla T_t(x)(T_t(x) - x) \rangle \, \rho_t dx + \int \langle T_t(x) - x, \nabla T_t \nabla \Phi_t \rangle \, \rho_t dx$$

$$- \int \langle T_t(x) - x, \nabla \Phi_t \rangle \, \rho_t dx + \frac{t}{2} \int \langle \nabla \Phi_t, \nabla T_t \nabla \Phi_t \rangle \, \rho_t dx$$

$$- \frac{1}{2} \int \langle T_t(x) - x, \nabla \Phi_t \rangle \, \rho_t dx - \frac{t}{2} \int \langle \partial_t \nabla \Phi_t, T_t(x) - x \rangle \, \rho_t dx$$

$$- \frac{t}{4} \int \langle T_t(x) - x, \nabla \|\nabla \Phi_t\|^2 \rangle \, \rho_t dx - \frac{t}{2} \int \langle \nabla \Phi_t, \nabla T_t \nabla \Phi_t \rangle \, \rho_t dx \tag{38}$$

$$+ \frac{t}{2} \int \|\nabla \Phi_t\|^2 \rho_t dx + \frac{t}{4} \int \|\nabla \Phi_t\|^2 \rho_t dx + \frac{t^2}{4} \int \langle \nabla \Phi_t, \partial_t \nabla \Phi_t \rangle \, \rho_t dx$$

$$+ \frac{t^2}{8} \int \langle \nabla \Phi_t, \nabla \|\nabla \Phi_t\|^2 \rangle \, \rho_t dx + \frac{t^2}{4} \int \left\langle \nabla \Phi_t, \nabla \frac{\delta E}{\delta \rho_t} \right\rangle \rho_t dx$$

$$- \frac{t}{2} \int \left\langle T_t(x) - x, \nabla \frac{\delta E}{\delta \rho_t} \right\rangle \rho_t dx.$$

The expression of (38) can be reformulated into

$$\dot{\mathcal{E}}(t) \leq - \frac{3}{2} \int \langle T_t(x) - x, \nabla \Phi_t \rangle \, \rho_t dx + \frac{3t}{4} \int \|\nabla \Phi_t\|^2 \rho_t dx$$

$$- \frac{t}{2} \int \left\langle T_t(x) - x, \partial_t \nabla \Phi_t + \frac{1}{2} \nabla \|\nabla \Phi_t\|^2 + \nabla \frac{\delta E}{\delta \rho_t} \right\rangle \rho_t dx$$

$$+ \frac{t^2}{4} \int \left\langle \nabla \Phi_t, \partial_t \nabla \Phi_t + \frac{1}{2} \nabla \|\nabla \Phi_t\|^2 + \nabla \frac{\delta E}{\delta \rho_t} \right\rangle \rho_t dx.$$

From (W-AIG) with $\alpha_t = 3/t$, we have the following equalities.

$$\frac{t^2}{4} \int \left\langle \nabla \Phi_t, \partial_t \nabla \Phi_t + \frac{1}{2} \nabla \|\nabla \Phi_t\|^2 + \nabla \frac{\delta E}{\delta \rho_t} \right\rangle \rho_t dx = - \frac{3t}{4} \int \|\nabla \Phi_t\|^2 \rho_t dx,$$

$$- \frac{t}{2} \int \left\langle T_t(x) - x, \partial_t \nabla \Phi_t + \frac{1}{2} \nabla \|\nabla \Phi_t\|^2 + \nabla \frac{\delta E}{\delta \rho_t} \right\rangle \rho_t dx = \frac{3}{2} \int \langle T_t(x) - x, \nabla \Phi_t \rangle \, \rho_t dx.$$

As a result, $\dot{\mathcal{E}}(t) \leq 0$. This completes the proof.

## C.4 COMPARISON WITH THE PROOF IN THE ACCELERATED FLOW

The accelerated flow in (Taghvaei & Mehta, 2019) is given by

$$\frac{dX_t}{dt} = e^{\alpha_t - \gamma_t} Y_t, \quad \frac{dY_t}{dt} = -e^{\alpha_t + \beta_t + \gamma_t} \nabla \left( \frac{\delta E}{\delta_{\rho_t}} \right)(X_t). \tag{39}$$

Here the target distribution satisies $\rho_\infty(x) = \rho^*(x) \propto \exp(-f(x))$. Suppose that we take $\alpha_t = \log p - \log t$, $\beta_t = p \log t + \log C$ and $\gamma_t = p \log t$. Here we specify $p = 2$ and $C = 1/4$. Then the accelerated flow (39) is identical to (W-AIG-P-KL) if we replace $Y_t$ by $2t^{-3}V_t$. The Lyapunov function in (Taghvaei & Mehta, 2019) follows

$$V(t) = \frac{1}{2}\mathbb{E}[\|X_t + e^{-\gamma_t}Y_t - T_{\rho_t}^{\rho^*}(X_t)\|^2] + e^{\beta_t}(E(\rho) - E(\rho^*))$$

$$= \frac{1}{2}\mathbb{E}[\|X_t + \frac{t}{2}V_t - T_{\rho_t}^{\rho^*}(X_t)\|^2] + \frac{t^2}{4}(E(\rho_t) - E(\rho^*))$$

$$= \frac{1}{2}\int \left\| -(T_t(x) - x) + \frac{t}{2}\nabla\Phi_t(x) \right\|^2 \rho_t(x)dx + \frac{t^2}{4}(E(\rho_t) - E(\rho^*)).$$

The last equality is based on the fact that $V_t = \nabla\Phi_t(X_t)$ and $T_t = T_{\rho_t}^{\rho^*}$ is the optimal transport plan from $\rho_t$ to $\rho^*$. This indicates that the Lyapunov function in (Taghvaei & Mehta, 2019) is identical to ours (8). The technical assumption in (Taghvaei & Mehta, 2019) follows

$$0 = \mathbb{E}\left[ \left( X_t + e^{-\gamma_t}Y_t - T_{\rho_t}^{\rho^*}(X_t) \right) \cdot \frac{d}{dt}T_{\rho_t}^{\rho^*}(X_t) \right]$$

$$= \mathbb{E}\left[ \left( X_t + \frac{t}{2}V_t - T_t(X_t) \right) \cdot \frac{d}{dt}T_t(X_t) \right]$$

$$= \mathbb{E}\left[ \left( X_t + \frac{t}{2}V_t - T_t(X_t) \right) \cdot ((\partial_t T_t)(X_t) + \nabla T_t V_t) \right]$$

$$= \int \left\langle x - T_t(x) + \frac{t}{2}\nabla\Phi_t(x), \partial_t T_t + \nabla T_t \nabla\Phi_t \right\rangle \rho_t dx$$

Based on $\partial_t T_t = -\nabla T_t u_t$ and Lemma 1, we have

$$\int \langle x - T_t(x), \partial_t T_t + \nabla T_t \nabla\Phi_t \rangle \rho_t dx = \int \langle x - T_t(x), \nabla T_t(\nabla\Phi_t - u_t) \rangle \rho_t dx = 0.$$

$$\int \langle \nabla\Phi_t, \partial_t T_t + \nabla T_t \nabla\Phi_t \rangle \rho_t dx = \int \langle \nabla\Phi_t, \nabla T_t(\nabla\Phi_t - u_t) \rangle \rho_t dx$$

$$= \int \langle \nabla\Phi_t - u_t, \nabla T_t(\nabla\Phi_t - u_t) \rangle \rho_t dx \geq 0.$$

As a result, we have

$$\mathbb{E}\left[ \left( X_t + e^{-\gamma_t}Y_t - T_{\rho_t}^{\rho_\infty}(X_t) \right) \cdot \frac{d}{dt}T_{\rho_t}^{\rho_\infty}(X_t) \right] = \frac{t}{2}\int \langle \nabla\Phi_t - u_t, \nabla T_t(\nabla\Phi_t - u_t) \rangle \rho_t dx \geq 0.$$

In 1-dimensional case, because $\nabla \cdot (\rho_t(u_t - \nabla\Phi_t)) = 0$ indicates that $\rho_t(u_t - \nabla\Phi_t) = 0$. For $\rho_t(x) > 0$, we have $u_t(x) - \nabla\Phi_t(x) = 0$. So the technical assumption holds. In general cases, although $u_t = \partial_t(T_t)^{-1} \circ T_t$ satisfies $\nabla \cdot (\rho_t(u_t - \nabla\Phi_t)) = 0$, but this does not necessary indicate that $u_t = \nabla\Phi_t$. Hence, $\mathbb{E}\left[ \left( X_t + e^{-\gamma_t}Y_t - T_{\rho_t}^{\rho_\infty}(X_t) \right) \cdot \frac{d}{dt}T_{\rho_t}^{\rho_\infty}(X_t) \right] = 0$ does not necessary hold except for 1-dimensional case.

## D  IMPLEMENTATION DETAILS IN THE NUMERICAL EXPERIMENTS

In this section, we elaborate on the implementation details in the numerical experiments.

### D.1  DETAILS IN SUBSECTION 6.1

The initial distribution of the particle system follows the standard Gaussian $\mathcal{N}(0, I)$. The objective density function is

$$\rho^*(x) \propto \exp(-2(\|x\| - 3)^2)(\exp(-2(x_1 - 3)^2) + \exp(-2(x_1 + 3)^2)), \quad x \in \mathbb{R}^2.$$

All methods run for 200 iterations using the same fixed step size $\tau = 0.1$.

## D.2  DETAILS IN SUBSECTION 6.2

The discretization the Wasserstein gradient flow in Gaussian follows

$$\Sigma_{k+1} = \Sigma_k - 2\tau(\Sigma_k \nabla_{\Sigma_k} E(\Sigma_k) + \nabla_{\Sigma_k} E(\Sigma_k)\Sigma_k). \tag{40}$$

The discretization of (W-AIG-G) is given by

$$S_{k+1} = \alpha_k S_k - \sqrt{\tau}(S_k^2 + 2\nabla_{\Sigma_k} E(\Sigma_k)), \quad \Sigma_{k+1} = (I + \sqrt{\tau}S_{k+1})\Sigma_k(I + \sqrt{\tau}S_{k+1}). \tag{41}$$

The choice of $\alpha_k$ is same as the one for (9), based on whether $\beta$ is known. To ensure the update of $\Sigma_{k+1}$ won't blow up, we check whether $\mathrm{tr}(\tau S^2) > n$. If this holds, we restart the algorithm. In fact, the update on $\Sigma_k$ can be viewed as the exponential map from $\Sigma_k$ with the direction $\Sigma_k S_{k+1} + S_{k+1}\Sigma_k$. Nevertheless, this is an exponential map if and only if $I + \sqrt{\tau}S_{k+1}$ is positive definite. The update rule of $\sqrt{\tau}S_{k+1}$ can rewrite into

$$\sqrt{\tau}S_{k+1} = \alpha_k\sqrt{\tau}S_k - (\sqrt{\tau}S_k)^2 - 2\tau\nabla_{\Sigma_k} E(\Sigma_k). \tag{42}$$

Because of the existence of $-(\sqrt{\tau}S_k)^2$, as long as the spectral radius (the eigenvalue with the largest absolute value) of $\sqrt{\tau}S_k$ is greater than 1, then $I + \sqrt{\tau}S_{k+1}$ cannot maintain to be positive definite. So at this time we need to restart the algorithm. Nevertheless, to compute the spectral radius is computational costly. Instead, we use a weaker condition $\mathrm{tr}(\tau S_k^2) > n$.

We also introduce the adaptive restart technique for (41). Consider

$$\varphi_k = -\mathrm{tr}(S_{k+1}\Sigma_k \nabla_{\Sigma_k} E(\Sigma_k)).$$

This restarting criterion corresponds to the particle version of (13). Similarly, if $\varphi_k < 0$, we restart the algorithm using initial values $\Sigma_0 = \Sigma_k$ and $S_0 = 0$.

For the particle implementation, the update rule of WGF writes

$$X_k^i = -\tau(\nabla f(X_k^i)) + \xi_k(X_k^i)),$$

where $\xi_k$ is computed based on KDE (11).

## D.3  DETAILS IN SUBSECTION 6.3

We follow the same setting as (Liu & Wang, 2016), which is also adopted by Liu et al. (2018; 2019). The data set is split into $80\%$ for training and $20\%$ for testing. The mini-batch size is taken as $50$. For MCMC, the number of particles is $N = 1000$; for other methods, the number of particles is $N = 100$. The BM method is not applied to SVGD in selecting the bandwidth.

The initial step sizes for the compared methods are given in Table 1. The step size of SVGD is adjusted by Adagrad, which is same as (Liu & Wang, 2016). For WNAG, AIG and WRes, the step size is give by $\tau_l = \tau_0/l^{0.9}$ for $l \geq 1$. The parameters for WNAG and Wnes are identical to (Liu et al., 2018) and (Liu et al., 2019). For MCMC, WGF and AIG-r, the step size is multiplied by $0.9$ every 100 iterations. We record the cpu-time for each method in Table 2. The computational cost

| Method | MCMC | SVGD | WNAG | Wnes | WGF | AIG | AIG-r |
|---|---|---|---|---|---|---|---|
| Step size $\tau$ | 1e-5 | 0.05 | 1e-6 | 1e-5 | 1e-5 | 1e-5 | 1e-6 |

Table 1: Initial step sizes of algorithms in comparison.

of the BM method is much higher than the MED method because we need to evaluate the MMD of two particle systems several times in optimizing the subproblem.

| Method | MCMC | SVGD | WNAG | Wnes | WGF | AIG | AIG-r |
|---|---|---|---|---|---|---|---|
| BM | 13.962 | 5.539 | 78.038 | 78.509 | 79.006 | 79.094 | 78.945 |
| MED | 13.909 | 5.581 | 5.623 | 5.625 | 5.395 | 5.890 | 5.689 |

Table 2: Averaged cpu time(s) cost for algorithms in comparison.

