# OpenReview forum: "Accelerated Information Gradient flow"
_ICLR.cc/2020/Conference — Reject_

### Official Review · AnonReviewer2 · 2019-10-24
**Official Blind Review #2**

**Rating:** 3

**Review:**

The paper attempts to develop a counterpart of the well-known Nesterov accelerated gradient method for gradient flows on the space of probability measures equipped with an information metric. This is an important problem which is useful for optimization on probability spaces. The accelerated gradient flow is developed by leveraging a damping Hamiltonian flow. The paper focuses mainly on the case with the Wasserstein metric and provides a convergence analysis. Practical considerations such as discretizing the accelerated flow and bandwidth selection are developed for the use of the method in practical problems.

Although the paper has some important merit, I find the paper extremely hard to follow, partly because of its writing style. There is not enough motivation and explanation for the ideas presented. Some discussions and sentences either do not make much sense to me or read badly. For example, this sentence "For the Wasserstein gradient, many classical methods such as Markov Chain Monte Carlo .... are based on this framework..." doesn't make sense, as the development of MCMC is never based on Wasserstein gradient. Or the sentence right before it "For the Fisher-Rao gradient, classical results including Adam ... and K-FAC .. demonstrate its effectiveness in ...": it's not clear what the authors are trying to say here. Adam is not relevant to the Fisher-Rao natural gradient while K-FAC is just an approximation method and isn't a good reference for demonstrating the effectiveness of the natural gradient. Also, there are many English typos and grammar errors.

I didn't read the proof carefully due to the time constraint, so I cannot judge on the theoretical part of the paper. The numerical experiment is quite limited as it considers very simple problems (a toy example, a single Gaussian distribution and a logistic regression problem). As such, I think there isn't enough evidence to judge the usefulness of the proposed method in practice.

Having said that, I believe this paper can be an important contribution if the authors invest more time on refining its presentation and if more thorough experimental studies are conducted.


**Experience Assessment:**

I have read many papers in this area.

**Review Assessment: Checking Correctness Of Derivations And Theory:**

I assessed the sensibility of the derivations and theory.

**Review Assessment: Checking Correctness Of Experiments:**

I assessed the sensibility of the experiments.

**Review Assessment: Thoroughness In Paper Reading:**

I read the paper at least twice and used my best judgement in assessing the paper.

---

> ### Author Response · Authors · 2019-11-09
> **Summary: AnonReviewer2 skips theoretical parts of the paper and is not familiar with the area.**
>
> We make several clarifications for the presentation in our paper.
>
> 1. Because of the ten pages of limitations in ICLR, we present the paper focused mainly on the result. In the revision, we will provide more instructions and derivations for the proposed flow formulation.
>
> 2. In the sentence "For the Wasserstein gradient, many classical methods such as Markov Chain Monte Carlo .... are based on this framework...", we particularly emphasize on Langevin MCMC, which is the discretization of overdamped Langevin diffusion. The probability density of overdamped Langevin diffusion is the gradient flow of KL divergence concerning the Wasserstein metric. This property has been known for many ML communities and papers, see [1].
>
> 3. For the sentence "For the Fisher-Rao gradient, classical results including Adam ... and K-FAC .. demonstrate its effectiveness in ...". The authors of Adam in Section 5 of [2] pointed out that the bias-corrected second raw moment estimate $\hat v_t$ is an approximation to the diagonal of the Fisher information matrix (actually it is the empirical Fisher matrix, see Section 4 of [3]). K-FAC utilizes the block-diagonal approximation of the Fisher Information matrix. Adam and K-FAC all utilize the curvature information from the Fisher-Rao metric, although they use different kinds of approximation.
>
> 4. For the numerical part, we have shown that our theory works for successful cases, including high dimensional Gaussian and Logistic regression problems. They confirm our theoretical justifications shown in this paper. Here we point out the AIG flow is stiff. A particular numerical time discretization and optimization method, such restart technique, needs to be developed in this framework. In the future, we will also conduct more extensive numerical tests for our proposed methods.
>
> [1]: Cheng and Bartlett: Convergence of Langevin MCMC in KL-divergence, 2017
>
> [2]: Kingma and Ba, Adam: A Method for Stochastic Optimization, 2015.
>
> [3]: Pascanu and Bengio, Revisiting natural gradient for deep networks, 2013.

---

### Official Review · AnonReviewer3 · 2019-10-26
**Official Blind Review #3**

**Rating:** 3

**Review:**

I acknowledge reading the rebuttal of the authors. Thank you for your clarifications and explanation. My point was this paper would make a good submission to ICLR if it was better motivated presented and explained to a wider audience. Unfortunately in its current form it can only reach a limited audience.

####
Summary of the paper:

The paper proposes accelerated information flows under Wasserstein or Fisher rao metric . i.e a method for solving minimization of probability functional , where the probability space is either endowed with the Wasserstein or the Fisher distance.

Gradient descent methods in euclidian spaces can be accelerated using a type of momentum , this paper extends this to gradient flows, using similar formalism of Hamiltonian that appeals to the dynamic of a the particles and the velocity (or momentum, $H(x,p)=\frac{1}{2}||p||^2+ \mathcal{E}(x)$).  Writing down the lagrangian one obtains two co-evolving PDE one of the dynamic of the density and one for the potential . The PDEs are specified for both Fisher Rao, and Wasserstein distance. The hamiltonian for example for the Wasserstein distance $H(\rho_t,\Phi_t)=\frac{1}{2}\int||\nabla_x \Phi_t||^2+ \mathcal{E}(\rho_t)$.  and PDEs amounts a continuity equation for the density  evolving with drift $\nabla \Phi_t$, the evolution of the momentum $\Phi_t$ is also given by a PDE.

Proposition 2 of the paper gives the particles differential equation corresponding to the system of PDEs. For the energy being the KL divergence an explicit expression is given , this expression remains difficult in practice since it needs the knowledge of the density $\rho_t$. Authors propose in the application section to use gaussian approximation , or using a kernel density estimators. The Bandwidth of the kernel is then choosen using a heuristic proposed in the paper.

The paper then focuses on deriving expression for flows when the densities are centered gaussians, and this amounts to an ODE on the covariance , the ODE is discretized in Appendix D.2 to lead to  computational method. Then convergence of the flow is analyzed for the wasserstein accelerated flows, under "\beta- convextiy" in the wasserstein sense of the functional.

Some experiments of the particle based method are shown on synthetic experiments and in bayesian logistic regression.

Review:

Contribution/ Clarity:

The main contribution of the paper is in deriving the accelerated gradient flow for the wasserstein distance this was also addressed in a recent paper [Taghevia and Mettha 2019].

The technical contribution is interesting but given that this field of flows in probability space is still not very well spread in the ML community, I wonder if ICLR is the best fit for this type of work.  I support good theoretical work, but I think the authors could have done a better job in exposing the ideas how they extend form euclidean space, to manifolds, to probability spaces gradually. Simple derivations of euclidean space Hamiltonian will help the reader that is not exposed to such literature. I think the paper will benefit from a less technical writing in introducing the ideas coming from euclidean space and in conveying the intuitions.

Comments:

- In the proof of Proposition 2 you give the expression of evolution of $dV_t$ by conservation of the momentum. Could you please elaborate more how you obtain this expression, and where you proved the conservation of momentum?

- In term of damping if ones uses the Wasserstein Fisher Rao flows  , one obtain also accelerartion , maybe you can comment on that ? since you analyze both flows , would be interesting to discuss the relation to Global convergence of neuron birth-death dynamics, that shows that an acceleration is obtained via WFR flows, since it will introduce a damping as well.

- since MCMC and BM method lead to similar result what is the advantage of the wasserstein accelerated flow? one could also implement also an accelerated langevin dynamic




**Experience Assessment:**

I have published one or two papers in this area.

**Review Assessment: Checking Correctness Of Derivations And Theory:**

I assessed the sensibility of the derivations and theory.

**Review Assessment: Checking Correctness Of Experiments:**

I carefully checked the experiments.

**Review Assessment: Thoroughness In Paper Reading:**

I read the paper at least twice and used my best judgement in assessing the paper.

---

> ### Author Response · Authors · 2019-11-09
> **Summary of the response: If ICLR is not the best fit for this paper, would you please suggest us some other leading learning conferences or journals for this theoretical work?**
>
> Thanks for your comments.
>
> 1. We agree that the calculus or flows in the space of probability measures, studied in optimal transport, mean-field games and information geometry, has not been widely known in the ML community yet. The purpose of this paper is to show that they could be useful in designing algorithms for ML tasks, especially for those with sampling-efficient properties. We expect these new sampling methods would interact with more in-depth optimization techniques.
>
> 2. For explaining ideas, we agree that we could improve the presentation.
> In the future version, we would like to present the damped Hamiltonian flow in the Euclidean space first, then Riemannian manifold and the probability space. We want to make more comparisons between the Euclidean case and the case in the probability space to convey the idea more explicitly and write them in a better form. We would present a systemic study of various metrics in probability measures, including Fisher, Wasserstein, Wasserstein-Fisher-Rao, or modified Wasserstein metric.
>
> 3.For the contribution, we emphasize that [1], i.e. [Taghevia and Mettha 2019] in ICML, introduced the Accelerated Wasserstein gradient flows.
>
> We summarize our formulation as follows:
>
> Firstly, our work is to give a systemic derivation for all related metrics in probability measures. Fisher-Rao and Wasserstein metric are well-known examples. In a revision, we would like to provide many examples, including various machine learning communities concerned metrics, such as modified optimal transport metric, in probability space.
>
> Secondly, focus on the Accelerated Wasserstein gradient flows,
> we study their theoretical behaviors in this paper. We prove the existence of flow in Gaussian cases and show the improved convergence properties for general measures. In the proof of Lemma 1 in Appendix C.3, we discover the fact that the Hodge decomposition in optimal transport provides an additional acceleration factor. In other words, the larger the dimension, the better is the acceleration constant. This is a unique property for optimal transport-like metrics, coming from fluid dynamics, which is a new observation that [1] has not discovered.
>
> Thirdly and most importantly, the time discretization or the choice of step size is very crucial for the performance of the algorithm. We believe that by writing this calculus more clearly, it would help us to derive more practical computational effects. In our paper, we do observe that the AIG flows are very stiff systems. Careful and thoughtful time discretizations are key factors for the acceleration properties, which require deeper understanding in optimization and machine learning. For this reason, we bring two algorithms here, such as the Brownian motion method and the restart technique. Because of the ten pages limitation, the current formulation is not well explained. We would explain this more clearly in a future revision.

---

> ### Author Response · Authors · 2019-11-09
> **Response to the technical comments**
>
> We also answer the technical comments as follows.
>
> 1. For the conservation of momentum, this step in proving Proposition 2 is based on the concept of material derivative in the fluid dynamics: given $\frac{d}{dt}X_t= \nabla \Phi_t(X_t)$, for any (vector-valued) function $f(x,t)$, the material derivation $\left(\frac{D}{Dt} f\right)(X_t,t) = \partial_t\left( f(X_t,t)\right)$ is defined by
> $$
> \frac{D}{Dt} f(X_t,t) = \nabla \Phi_t \cdot \nabla_x f(X_t,t)+\partial_t f(X_t,t).
> $$
>
> 2. The Wasserstein Fisher-Rao metric (WFR) and neuron birth-death dynamics are also of great interest for us to study. We want to establish similar results for the acceleration of the WFR metric. This requires a systemic study of this metric. In these studies, the acceleration effect in Lemma 1 in Appendix C.3 needs to be carefully conducted in the future.
> Potential collaborations are welcomed.
>
> 3. MCMC and the BM method do lead to similar results for the Wasserstein gradient flow. These methods, in the sense of probability, are first-order methods. In our paper, we numerically verify the theory that our accelerated first order method behaves better. For example, the order of convergence becomes $O(1/k^2)$, not $O(1/k)$.
>
> Here consider
> $$
> \ddot x + a \dot x + \nabla f + \sqrt{2} B_t = 0, \quad (1)
> $$
>
> $$
> \ddot x+ a \dot x + \nabla f+ \nabla \log \rho_t  =0. \quad   (2)
> $$
>
> One can prove that the classical accelerated Langevin dynamic (1) is not the first-order method in the sense of probability distributions. In other words, we cannot use the Brownian motion in the accelerated flows because $\rho_t$ is the density of marginal distribution on $X_t$, and the Brownian motion leads to the logarithm of the density of joint distribution of $(X_t,V_t)$. It is worth mentioning that this viewpoint has also been discussed in detail in Appendix D of [1]. For the more systemic study of this difference, we recommend interested readers to see Wasserstein Hamiltonian flows in [2].
>
> [1]: Taghvaei and Mehta, Accelerated Flow for Probability Distributions, ICML, 2019.
>
> [2]: Chow, Li, and Zhou, Wasserstein Hamiltonian flows, Journal of differential equations, 2019.

---

### Decision · Program_Chairs · 2019-12-19

**Decision:**

Reject

**Comment:**

The paper makes its contribution by deriving an accelerated gradient flow for the Wasserstein distances. It is technically strong and demonstrates it applicability using examples fo Gaussian distributions and logistic regression.

Reviewer 3 provided a deep technical assessment, pointing out the relevance to our ML community since these ideas are not yet widespread, but had concerns about the clarity of the paper. Reviewer 2 had similar concerns about clarity, and was also positive about its relevance to the ML community. The authors provided details responses to the technical questions posed by the reviewers. The AC believes that such work is a good fit for the conference. The reviewers felts that this paper does not yet achieve the aim of making this work more widespread and needs more focus on communication.

This is a strong paper and the authors are encouraged to address the accessibility questions. We hope the review offers useful points of feedback for their future work.